# ACCUMULATOR-AWARE POST-TRAINING QUANTIZATION FOR LARGE LANGUAGE MODELS

## ABSTRACT

Several recent studies have investigated low-precision accumulation, reporting improvements in throughput, power, and area across various platforms. However, the accompanying proposals have only considered the quantization-aware training (QAT) paradigm, in which models are fine-tuned or trained from scratch with quantization in the loop. As models continue to grow in size, QAT techniques become increasingly more expensive, which has motivated the recent surge in post-training quantization (PTQ) research. To the best of our knowledge, ours marks the first formal study of accumulator-aware quantization in the PTQ setting. To bridge this gap, we introduce AXE—a practical, low-overhead framework of accumulator-aware extensions designed to endow overflow avoidance guarantees to existing layer-wise PTQ algorithms. We theoretically motivate AXE and demonstrate its flexibility by implementing it on top of two state-of-the-art PTQ algorithms: GPFQ and OPTQ. We further generalize AXE to support multi-stage accumulation for the first time, opening the door for full datapath optimization and scaling to large language models (LLMs). We evaluate AXE across autoregressive language generation models, and observe significant improvements in the trade-off between accumulator bit width and model accuracy over baseline methods.

## 1 INTRODUCTION

Modern deep learning models have scaled to use billions of parameters, requiring billions (or even trillions) of multiply-accumulate (MAC) operations during inference. Their enormous size presents a major obstacle to their deployment as their compute and memory requirements during inference often exceed the budgets of real-world applications. As a result, model compression has emerged as an important active area in deep learning research, with quantization being among the most prevalent techniques studied and applied in practice (Wu et al., 2020; Nagel et al., 2021; Gholami et al., 2022).

Quantization techniques commonly reduce inference costs by restricting the precision of its weights and activations. Although substituting the standard 32-bit floating-point operands for low-precision counterparts can drastically reduce the cost of multiplications, this only accounts for part of the core MAC operation; the resulting products are often still accumulated at 32 bits. Recent studies have demonstrated that also restricting the precision of the accumulator can yield significant benefits (see Section 2.2). However, exploiting such an optimization is non-trivial in practice as reducing the width of the accumulator exponentially increases the risk of numerical overflow, which is known to introduce arithmetic errors that significantly degrade model accuracy (Ni et al., 2020).

To address this, recent work has proposed an accumulator-aware quantization paradigm that entirely eliminates the risk of numerical overflow via strict learning constraints informed by theoretical guarantees (Colbert et al., 2023). The resulting scope of investigations has been limited to the quantization-aware training (QAT) setting in which models are trained from scratch or fine-tuned from checkpoints with quantization in the loop (Colbert et al., 2023; 2024). With the rising training costs of modern deep learning models (*e.g.*, large language models), it is important to develop methods that are equally as effective in the post-training quantization (PTQ) setting, where pre-trained models are directly quantized and calibrated using relatively modest resources. However, controlling the accumulator bit width in such a scenario is non-trivial. In this work, we characterize and address these challenges, introduce a practical framework for their investigation, and establish a new state-of-the-art for accumulator-aware weight quantization in the PTQ setting.

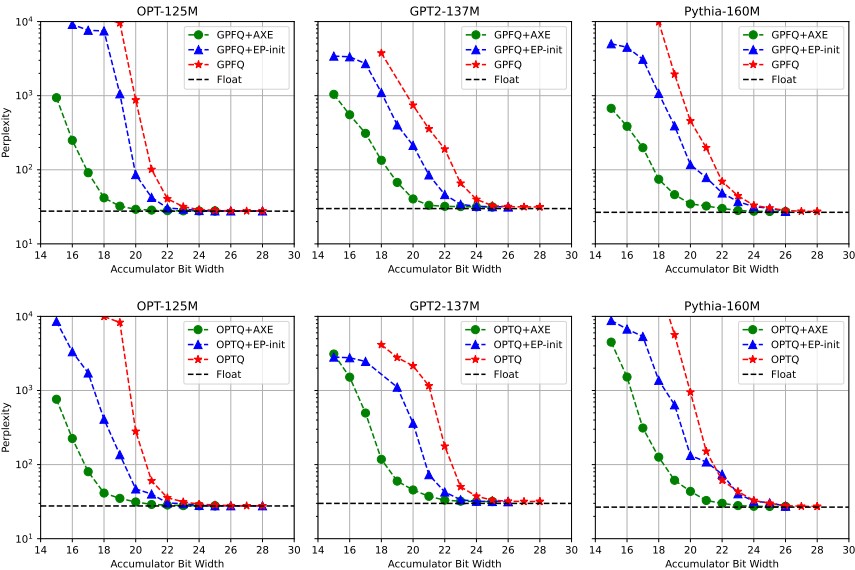

Figure 1: To reduce the minimum accumulator bit width required to avoid overflow during inference, one could naïvely manipulate the weight and activation bit widths according to the data type bound derived by Colbert et al. (2023). To date, Euclidean projection-based initialization (EP-init) (Colbert et al., 2024) serves as the best alternative to this approach, but, as the name suggests, it has only been studied as a QAT initialization strategy. AXE (**green circles**) significantly improves the trade-off between accumulator bit width and model quality for language models evaluated on WikiText2 (Merity et al., 2016) when compared to EP-init (**blue triangles**) and naïve bit width manipulation (**red stars**) for both GPFQ (Lybrand & Saab, 2021) (top) and OPTQ (Frantar et al., 2022) (bottom).

**Contribution.** We introduce AXE, a framework of accumulator-aware extensions designed to endow overflow avoidance guarantees to any layer-wise PTQ algorithm that greedily quantizes weights one at a time, provided the base algorithm is also amenable to activation quantization. We theoretically motivate AXE and demonstrate its flexibility by presenting accumulator-aware variants of both GPFQ and OPTQ. We evaluate AXE across pre-trained language generation models and show that it significantly improves the trade-off between accumulator bit width and model quality when compared to baseline methods. We visualize this trade-off using the Pareto frontiers in Figure 1, which provide the minimum observed perplexity for a given target accumulator bit width. Furthermore, unlike prior accumulator-aware quantization methods, which assume a monolithic accumulator, we generalize AXE to support multi-stage accumulation, which enables accumulator-aware quantization of large language models (LLMs) for the first time. Indeed, our results show that AXE scales extremely well to billion-parameter language models when targeting multi-stage accumulation, supporting the scaling hypothesis proposed by Colbert et al. (2024).

## 2 PRELIMINARIES

We denote the $K_l$-dimensional input activations to layer $l$ as $\boldsymbol{x}^{(l)} \in \mathbb{R}^{K_l}$, where $\boldsymbol{X}^{(l)} \in \mathbb{R}^{K_l \times D}$ denotes a matrix of $D$ such inputs. The weight matrix for layer $l$ with $K_l$ input neurons and $C_l$ output neurons is similarly denoted as $\boldsymbol{W}^{(l)} \in \mathbb{R}^{C_l \times K_l}$; its quantized counterpart is denoted as $\boldsymbol{Q}^{(l)} \in \mathcal{A}_M^{C_l \times K_l}$, where we use $\mathcal{A}_b^{m \times n}$ to denote the space of all $m \times n$ matrices whose elements are part of a fixed $b$-bit alphabet defined by the target quantization space. For example, the alphabet of signed $b$-bit integers is $\mathcal{A}_b := \{k : -2^{b-1} + 1 \leq k \leq 2^{b-1} - 1, k \in \mathbb{Z}\}$, assuming a sign-magnitude representation, where $\mathbb{Z}$ is the space of all scalar integers.

For layer $l$, our notation yields $C_l$ independent dot products of depth $K_l$ for each of the $D$ inputs. For clarity, and without loss of generality, we often assume $C_l = 1$ when focusing on a single layer $l$ so that we can use $\boldsymbol{w}^{(l)}$ to denote the weight matrix for layer $l$. When dropping their superscript, $\boldsymbol{x}$ and $\boldsymbol{w}$ denote generic inputs and weights in $\mathbb{R}^K$, and $\tilde{\boldsymbol{x}}$ and $\boldsymbol{q}$ denote their quantized counterparts.

## 2.1 POST-TRAINING QUANTIZATION

Standard quantization operators, referred to as quantizers, are commonly parameterized by zero-point $z$ and strictly positive scaling factor $s$, as shown in Eq. 1 for weight tensor $\boldsymbol{w}$. Our work focuses on uniform integer quantization, where $z$ is an integer value that ensures that zero is exactly represented in the quantized domain, and $s$ is a strictly positive scalar that corresponds to the resolution of the quantizer. Scaled values are commonly rounded to the nearest integer, denoted by $\lceil \cdot \rfloor$, and elements that exceed the representation range of the quantized domain $\mathcal{A}_b$ are clipped.

$$\mathcal{Q}(\boldsymbol{w}) := s \cdot \left( \text{clip}\left( \left\lceil \frac{\boldsymbol{w}}{s} \right\rfloor + z; \min \mathcal{A}_b, \max \mathcal{A}_b \right) - z \right) \tag{1}$$

Methods for tuning these quantizers broadly fall into two paradigms: quantization-aware training (QAT) and post-training quantization (PTQ). QAT methods train or fine-tune a neural network with quantization in the loop, which often requires significant compute resources and sufficiently large datasets. Our work focuses on PTQ methods, which directly cast and calibrate pre-trained models and often rely on little to no data without end-to-end training. PTQ methods tend to follow a similar general structure, greedily casting and calibrating quantized models layer-by-layer or block-by-block while seeking to approximate the minimizer of the reconstruction error

$$\boldsymbol{q}^* = \arg\min_{\boldsymbol{q}} \frac{1}{2} \|\boldsymbol{X}^T \boldsymbol{w} - \tilde{\boldsymbol{X}}^T \boldsymbol{q}\|_2^2 \tag{2}$$

where $\boldsymbol{q}^*$ is the optimal quantized weights and $\tilde{\boldsymbol{X}}$ is the quantized counterpart of $\boldsymbol{X}$. Recent PTQ methods concentrate on "weight-only quantization", where $\tilde{\boldsymbol{X}} = \boldsymbol{X}$, to solely minimize memory storage and transfer costs (Lybrand & Saab, 2021; Frantar et al., 2022), and for good reason—the ever-increasing weight volume of state-of-the-art neural networks has rendered many hyper-scale transformer models memory-bound (Zhang et al., 2022a; Biderman et al., 2023). In such a scenario, weight-only quantization algorithms can better preserve model quality and still realize end-to-end throughput gains just by reducing data transfer costs, even with high-precision computations (usually FP16) (Frantar et al., 2022; Tseng et al., 2024). However, weight-only quantization provides limited opportunity to accelerate compute-intensive operations such as matrix multiplications, which is the focus of this work. Thus, we investigate methods that are amenable to *quantizing both weights and activations* to low-precision integers, which can realize throughput gains from both accelerated computation and reduced data traffic (Xiao et al., 2023; Li et al., 2024).

## 2.2 LOW-PRECISION ACCUMULATION

The majority of neural network quantization research targeting compute acceleration emphasizes low-precision weights and activations. While this can significantly reduce the costs of multiplications, the resulting products are often still accumulated using high-precision additions. As lower precision integer representations continue to increase in popularity (Dettmers & Zettlemoyer, 2023; Ma et al., 2024), one can expect a focus skewed towards weight and activation quantization to yield diminishing returns as high-precision additions can bottleneck throughput, power, and area (Ni et al., 2020; de Bruin et al., 2020; Xie et al., 2021; Colbert et al., 2024). For example, Ni et al. (2020) show that when constraining weights and activations to 3-bit $\times$ 1-bit multipliers, the cost of 32-bit accumulation consumes nearly 75% of the total power and 90% of the total area of their custom MAC unit; they report up to $4\times$ power savings and $5\times$ area reduction when reducing to 8-bit accumulation.

Reducing the accumulator bit width is non-trivial in practice as it exponentially increases the risk of numerical overflow, often introducing arithmetic errors that degrade model accuracy (Ni et al., 2020; Colbert et al., 2023). Existing methods to prepare quantized neural networks (QNNs) for low-precision accumulation often aim to either reduce the risk of numerical overflow (Xie et al., 2021; Li et al., 2022; Azamat et al., 2022) or mitigate its impact on model accuracy (Ni et al., 2020; Sakr et al., 2019; Blumenfeld et al., 2024). These empirical approaches rely on several assumptions that limit their real-world applicability. For one, empirical estimates of overflow rely on *a priori* knowledge of the input distribution, which is impractical to assume in many real-world scenarios and can even introduce vulnerabilities (Baier et al., 2019). Furthermore, overflow behavior can vary across platforms and programming languages, so designing methods to mitigate the detrimental impact of one particular overflow behavior (*e.g.*, wraparound two's complement arithmetic) limits portability across applications and accelerators. Finally, empirical approaches are unable to support applications that require guaranteed arithmetic correctness, such as encrypted inference (Lou & Jiang, 2019;

Stoian et al., 2023), and are known to break down when overflows occur too frequently (Ni et al., 2020). To address these concerns, recent work has proposed to avoid overflow altogether using accumulator-aware quantization (A2Q) (Colbert et al., 2023; 2024).

## 2.3 ACCUMULATOR-AWARE QUANTIZATION

Let $P^*$ denote the minimum accumulator bit width required to guarantee overflow avoidance for a given dot product. Aside from universally fixing the accumulator at 32 bits (or any other arbitrary maximum number of bits imposed by a given processor), the most conservative method to calculate $P^*$ considers the data types of the dot product operands, *i.e.*, weights and activations. Given inputs $\tilde{x} \in \mathcal{A}_N^K$ and weights $q \in \mathcal{A}_M^K$, $P^*$ is given by Eq. 3 as derived in Colbert et al. (2023), where $\mathbb{1}_{\text{signed}}(\tilde{x})$ is an indicator function that returns 1 if $\tilde{x}$ is signed and 0 otherwise.

$$P^* = \left\lceil \log_2 \left( 2^{\log_2(K)+N+M-1-\mathbb{1}_{\text{signed}}(\tilde{x})} + 1 \right) + 1 \right\rceil \tag{3}$$

Note that $P^*$ increases linearly with the bit widths of the operands and logarithmically with the depth of the dot product. Thus, for a fixed neural architecture, one could heuristically manipulate the weight and activation bit widths according to Eq. 3 to reduce $P^*$. However, the quantization design space ultimately limits the minimum attainable accumulator bit width, as well as the maximum attainable model accuracy for any target accumulator bit width (Colbert et al., 2023; 2024).

Colbert et al. (2024) show that one can directly target the accumulator bit width as an independent dimension of the quantization design space while still theoretically guaranteeing overflow avoidance. When accumulating $\tilde{x}^T q$ into a signed $P$-bit accumulator, one need only constrain $\|q\|_1$ according to Eq. 4, assuming that $\sum_i q_i = 0$.

$$\|q\|_1 \leq \frac{2^P - 2}{2^N - 1} \tag{4}$$

Motivated by this result, accumulator-aware QAT methods avoid overflow by constraining the $\ell_1$-norm of weights during training to ultimately restrict the range of dot product outputs during inference (Colbert et al., 2023; 2024). These approaches rely on weight normalization-based quantizers infused with strict accumulator-aware learning constraints. Although these approaches have yielded promising results in low-precision accumulation scenarios, the scope of their success is limited to the QAT setting (Colbert et al., 2023; 2024). However, from this family of QAT methods, one can apply the Euclidean projection-based initialization strategy (EP-init) (Colbert et al., 2024) to the PTQ setting without modification. However, we find that EP-init has two shortcomings in the PTQ setting: (1) it universally relies on the round-to-zero rounding function to ensure that $|Q(w_i)| \leq |w_i|$ for all $i$ (Colbert et al., 2023; 2024); and (2) it is a vector-wise projection operation that is not amenable to error correction (see Appendix C.2). We address these shortcomings in this work.

## 3 ACCUMULATOR-AWARE POST-TRAINING QUANTIZATION

The standard problem for neural network quantization aims to map high-precision values (*e.g.*, 32-bit floating-point) to low-precision counterparts (*e.g.*, 4-bit scaled integers) while locally minimizing the discrepancy between the output of the original model and that of the compressed one, as formalized by Eq. 2 in Section 2.1. In the post-training quantization (PTQ) setting, one often assumes the quantizer parameters (*i.e.*, scaling factor $s$ and zero point $z$) are fixed and that the individual weights can move freely, as in Lybrand & Saab (2021); Frantar et al. (2022); Hubara et al. (2021). Building from this, we formalize accumulator-aware post-training quantization as a constrained variant of the standard reconstruction problem in which the optimal quantized weights $q^*$ minimize local quantization error while also satisfying a strict $\ell_1$-norm constraint, as defined below.

$$q^* = \arg\min_{q} \frac{1}{2} \|X^T w - \tilde{X}^T q\|_2^2 \quad \text{s.t.} \quad \|q\|_1 \leq Z \tag{5}$$

To approximately solve this accumulator-constrained reconstruction problem, we introduce AXE, a practical low-overhead framework of general accumulator-aware extensions that endow guaranteed overflow avoidance to layer-wise quantization algorithms that greedily assign bits element-by-element (*e.g.*, GPFQ and OPTQ). AXE is built on two accumulator-aware constraints: (1) a soft

global constraint that discourages the underlying algorithm from opportunistically selecting quantized weights with high magnitudes; and (2) a strict local constraint that greedily limits the range of each selected quantized weight while error is iteratively corrected. In its standard form, AXE applies these constraints per-channel (or per-neuron) so that each dot product in the network is guaranteed to independently avoid overflow. Furthermore, without violating our constraints, we generalize our framework to also support multi-stage accumulation in the form of tiled dot products by applying our constraints in finer granularities. We first theoretically justify our solution using GPFQ, then provide accumulator-aware variants of both GPFQ and OPTQ.

## 3.1 ACCUMULATOR CONSTRAINTS WITHOUT ZERO-CENTERING

Our goal is to provide a theoretical guarantee of overflow avoidance when accumulating the dot product of $q$ by any $\tilde{x} \in \mathcal{A}_N^K$ into a signed $P$-bit register. If $q$ is a zero-centered vector such that $\sum_i q_i = 0$, then it is sufficient to constrain $\|q\|_1$ to satisfy the upper bound given by Eq. 4 in order to guarantee overflow avoidance (see Section 2.3). However, enforcing such a zero-centering constraint on a vector of integers is non-trivial in practice. Rather than directly enforcing this constraint on $q$, A2Q+ (Colbert et al., 2024) enforces these constraints on its floating-point counterpart $w$ and relies on the symmetry of the quantizer and the round-to-zero operator to ensure that $|Q(w_i)| \leq |w_i|$ for all $i$. We detach our solution from these zero-centering, round-to-zero, and symmetrical constraints.

For any $\tilde{x} \in \mathcal{A}_N^K$, each element $\tilde{x}_i$ lies within the closed interval $[\mu, \nu]$ for all $i = \{1, \cdots, K\}$, and $\nu - \mu = 2^N - 1$. It follows that the maximizing vector, $u = \arg\max_{\tilde{x}} \tilde{x}^T q$, and the minimizing vector, $v = \arg\min_{\tilde{x}} \tilde{x}^T q$, are:

$$u_i = \begin{cases} \nu, & \text{where } q_i \geq 0 \\ \mu, & \text{where } q_i < 0 \end{cases} \qquad v_i = \begin{cases} \mu, & \text{where } q_i \geq 0 \\ \nu, & \text{where } q_i < 0 \end{cases} \tag{6}$$

Fundamentally, to avoid overflow when accumulating $\tilde{x}^T q$ into a $P$-bit register, the result needs to fall within the register's representation range for any $\tilde{x} \in \mathcal{A}_N^K$. Without loss of generality, we derive our algorithm assuming a sign-magnitude accumulator for clarity and conciseness. Thus, to safely use a signed $P$-bit accumulator without overflow, the following inequalities need to be satisfied:

$$u^T q \leq 2^{P-1} - 1 \tag{7}$$

$$-v^T q \leq 2^{P-1} - 1 \tag{8}$$

To avoid zero-centering, one could generalize the result derived in Colbert et al. (2024) such that the bound relies on a variable center, e.g., $\sum_i q_i = \epsilon$. However, such a solution would still rely on the round-to-zero constraint. Furthermore, it precludes the use of greedy sequential algorithms where $\epsilon$ would be just as difficult to enforce as zero-centering, i.e., $\epsilon = 0$. Thus, rather than constraining the center, we greedily constrain the boundaries, as further discussed in Section 3.2.

## 3.2 ACCUMULATOR-AWARE GPFQ

The greedy path following quantization (GPFQ) algorithm (Lybrand & Saab, 2021) approaches the standard quantization problem by traversing the neural network graph to sequentially quantize each element in each layer while iteratively correcting for quantization error. At the $l$-th layer, this is done by greedily selecting each element $q_i$ to minimize the squared distance between the running sum $\sum_{j=1}^i q_j \tilde{X}_j$ and its analog $\sum_{j=1}^i w_j X_j$ such that

$$q_i^{(l)} = \arg\min_{p \in \mathcal{A}_M} \left\| \sum_{j=1}^i w_j^{(l)} X_j^{(l)} - \sum_{j=1}^{i-1} q_j^{(l)} \tilde{X}_j^{(l)} - p \tilde{X}_i^{(l)} \right\|_2 \tag{9}$$

where $\tilde{X}_i^{(l)}$ denotes samples for the $i$-th input neuron to the $l$-th layer assuming the first $l-1$ layers are quantized, and $\mathcal{A}_M$ is an $M$-bit fixed alphabet defined by the target quantization space. This simplifies to the following iteration rule as derived in Lybrand & Saab (2021), where $u_0^{(l)} = 0$.

$$q_i^{(l)} = \mathcal{Q}\left( \frac{\langle \tilde{X}_i^{(l)}, u_{i-1}^{(l)} + w_i^{(l)} X_i^{(l)} \rangle}{\|\tilde{X}_i^{(l)}\|_2^2} \right) \tag{10}$$

$$u_i^{(l)} = u_{i-1}^{(l)} + w_i^{(l)} X_i^{(l)} - q_i^{(l)} \tilde{X}_i^{(l)} \tag{11}$$

To add accumulator-awareness, we introduce two constraints that are agnostic to the symmetry of the quantizer and rounding function while still guaranteeing overflow avoidance. First, we introduce a soft $\ell_1$-norm regularization penalty that discourages the underlying algorithm (*e.g.*, GPFQ) from opportunistically selecting weights with high magnitudes. Second, we introduce a strict constraint that greedily limits the range of $q_i$ as error is iteratively corrected. This strict constraint is recursively applied element-by-element to ensure that Eqs. 7 and 8 are independently satisfied, which ultimately guarantees that Eq. 4 is satisfied without requiring a zero-centering constraint.

**Soft $\ell_1$-norm regularization penalty.** By design, greedy sequential quantization algorithms (*e.g.*, GPFQ and OPTQ) opportunistically alter weights to correct for as much error as possible in each step, often yielding high-magnitude quantized weights. However, this is unfavorable in the accumulator-aware quantization setting as high-magnitude weights consume more of the $\ell_1$-norm budget allocated per-channel (see Eq. 4). To address this, we penalize high-magnitude weights during error correction. We build from the sparse GPFQ formulation proposed by Zhang et al. (2023) as given by Eq. 12; the solution is given by Eq. 13, where $\Pi_\lambda(\boldsymbol{x}) := \text{sign}(\boldsymbol{x})(|\boldsymbol{x}| - \lambda)_+, (\cdot)_+$ denotes the rectified linear unit (ReLU), and $\lambda > 0$ is an arbitrary tuneable regularization parameter.

$$
q_i^{(l)} = \underset{p \in \mathcal{A}_M}{\arg\min} \left( \frac{1}{2} \left\| \sum_{j=1}^{i} w_j^{(l)} \boldsymbol{X}_j^{(l)} - \sum_{j=1}^{i-1} q_j^{(l)} \tilde{\boldsymbol{X}}_j^{(l)} - p \tilde{\boldsymbol{X}}_i^{(l)} \right\|_2^2 + \lambda |p| \left\| \tilde{\boldsymbol{X}}_i^{(l-1)} \right\|_2^2 \right) \tag{12}
$$

$$
= \mathcal{Q} \circ \Pi_\lambda \left( \frac{\langle \tilde{\boldsymbol{X}}_i^{(l)}, u_{i-1}^{(l)} + w_i^{(l)} \boldsymbol{X}_i^{(l)} \rangle}{\| \tilde{\boldsymbol{X}}_i^{(l)} \|_2^2} \right) \tag{13}
$$

Noticeably, this formulation is amenable to leverage EP-init (Colbert et al., 2024), which takes the same functional form. Thus, we tune our selection of $\lambda$ to be the optimal Lagrangian scalar derived from the solution to the constrained convex optimization problem formulated by Eq. 14. Here, the objective is to find the optimal Euclidean projection of $\boldsymbol{w}$ onto the $\ell_1$ ball of radius $Z$, where $Z$ is the accumulator-aware $\ell_1$-norm target given, up to a scaling, by the upper bound in Eq. 4. Thus, $\boldsymbol{v}^*$ is the vector that minimizes the projection onto the boundary of our constrained set *before* quantization.

$$
\boldsymbol{v}^* = \min_{\boldsymbol{v}} \frac{1}{2} \| \boldsymbol{v} - \boldsymbol{w} \|_2^2 \quad \text{subject to} \quad \| \boldsymbol{v} \|_1 \leq Z \tag{14}
$$

Define $\rho$ as the number of non-zero elements in the optimal solution and $\boldsymbol{\mu}$ as the result of sorting $\boldsymbol{w}$ by magnitude in descending order. The Lagrange multiplier $\lambda$ associated with the solution to the optimization problem is given by

$$
\lambda = \frac{1}{\rho} \left( \sum_{i=1}^{\rho} \mu_i - Z \right), \tag{15}
$$

which can be interpreted as the average difference between our scaled accumulator-aware $\ell_1$-norm target and the magnitudes of all non-zero elements in the optimal Euclidean projection $\boldsymbol{v}^*$. We direct the reader to Colbert et al. (2024) and Duchi et al. (2008) for the associated proofs and derivations. It is important to note that because this projection is derived before quantization it cannot guarantee overflow avoidance on its own; both error correction and rounding errors may violate our constraint. However, we observe that it consistently yields improvements in model quality (see Appendix C.2).

**Strict accumulator-aware constraint.** For clarity, and without loss of generality, we motivate our strict accumulator-aware constraint using the special case where $\tilde{\boldsymbol{x}}$ is represented with unsigned integers such that $\mu = 0$ and $\nu = 2^N - 1$. Note that this setting is common when following activation functions with non-negative dynamic ranges (*e.g.*, ReLUs), or when an appropriate non-zero-valued zero-point is adopted (*i.e.*, asymmetric quantization) (Gholami et al., 2022; Zhang et al., 2022b).

Let $\alpha$ denote the sum of all negative elements in $\boldsymbol{q}$, and let $\beta$ denote the sum of all positive elements in $\boldsymbol{q}$. From Eq. 7, we can derive the upper bound on $\beta$ given by Eq. 16, which can similarly be derived for $-\alpha$ from Eq. 8 in the case of sign-magnitude representations. Indeed, $\boldsymbol{u}^T \boldsymbol{q} \leq 2^{P-1} - 1$ is guaranteed whenever $\beta\nu + \alpha\mu \leq 2^{P-1} - 1$, which holds in the case of unsigned activations if

$$
\beta \leq \frac{2^{P-1} - 1}{2^N - 1}. \tag{16}
$$

Therefore, to quantize layer $l$ for a target $P$-bit accumulator, we introduce a practical mechanism to control the range of the dot product based on the following modified GPFQ scheme:

$$q_i^{(l)} = \mathcal{Q} \circ \Psi_{a_{i-1}^{(l)}, b_{i-1}^{(l)}} \circ \Pi_\lambda \left( \frac{\langle \tilde{\boldsymbol{X}}_i^{(l)}, u_{i-1}^{(l)} + w_i^{(l)} \boldsymbol{X}_i^{(l)} \rangle}{\|\tilde{\boldsymbol{X}}_i^{(l)}\|_2^2} \right) \tag{17}$$

$$a_i^{(l)} = A^{(l)} - \alpha_i \tag{18}$$

$$b_i^{(l)} = B^{(l)} - \beta_i \tag{19}$$

where $\alpha_i$ denotes the sum of all negative elements in $\boldsymbol{q}$ whose index is less than $i$ and $\beta_i$ is its positive counterpart, $A^{(l)}$ and $B^{(l)}$ (defined in Eq. 20) are respectively the upper limits of $\alpha_i$ and $\beta_i$, and the closed interval $[a_i^{(l)}, b_i^{(l)}]$ is the range greedily enforced on $q_i$ as error is iteratively corrected. We use $\Psi_{a,b}$ to denote the clipping function $\Psi_{a,b}(x) := \text{clip}(x; a, b)$, which is effectively a no-op when the range $[a_i^{(l)}, b_i^{(l)}]$ exceeds that of the quantized domain $\mathcal{A}_M$. This has the desired feature of being functionally equivalent to GPFQ when the accumulator is large enough (*e.g.*, 32 bits). Finally, recall that $u_i^{(l)}$ is given by Eq. 11 with $u_0^{(l)} = 0$, which remains unchanged.

By independently constraining the limits of $q_i$, our accumulator-aware variant avoids overflow without explicit zero-centering. To ensure rounding errors do not compromise our bounds, we use

$$-A^{(l)} = B^{(l)} = \frac{2^{P-1} - 1}{2^N - 1} - \max(\Delta) \tag{20}$$

where $\max(\Delta)$ denotes the worst-case difference in raw magnitude caused by rounding; for example, $\max(\Delta) = 0.5$ for round-to-nearest and $\max(\Delta) = 0$ for round-to-zero. Thus, our formulation and resulting iteration rules are also agnostic to the symmetry of the quantizer and its choice of rounding function. We also note that, while our derivation considers the sign-magnitude representation for the clarity that its symmetry provides, the separate consideration of $A^{(l)}$ and $B^{(l)}$ is useful for asymmetric representations (*e.g.*, two's complement).

### 3.3 AXE: Accumulator-Aware Extensions

While the theoretical justification we presented is tied to the formulation of GPFQ and its derivations, we can extract our constraints to construct a generalized framework that enables the investigation of accumulator-aware PTQ with any iterative algorithm that sequentially assigns bits, assuming the algorithm is also amenable to activation quantization. Our framework consists of two steps: (1) accumulator-aware projection based on our soft $\ell_1$-norm regularization penalty; and (2) greedy accumulator-aware clipping based on our strict range limits. We further generalize AXE to support multi-stage accumulation, which has implications for tiled dot products and SIMD parallelization.

In Appendix A, we present the pseudocode for our accumulator-aware variants of GPFQ and OPTQ. In both cases, $\lambda$ is derived per-channel before quantization. Adjusted weight values are greedily projected on the $\ell_1$ ball accordingly, then clipped to the difference between the cumulative sum of positive and negative elements and their respective limits. The resulting set of quantized weights $\boldsymbol{Q} \in \mathcal{A}_M^{K \times C}$ is then guaranteed to avoid overflow when accumulating its inner product with any $\tilde{\boldsymbol{X}} \in \mathcal{A}_N^{K \times D}$ into $P$-bit signed registers. Unlike the base GPFQ and OPTQ algorithms, our accumulator-aware variants *require* quantized activations to calculate the accumulator-aware limits in Eq. 20.

**Multi-Stage Accumulation.** Our accumulator-aware constraints can be generalized to target customized datapaths beyond user-specific accumulator bit widths. Unlike A2Q and A2Q+, which assume a monolithic accumulator for each dot product (Colbert et al., 2023; 2024), we generalize our framework to support multi-staged accumulation as visualized in Figure 2. In such a scenario, our constraints are enforced on the quantized weights in tiles of size $T$ so that each partial dot product can be concurrently computed by an atomic MAC unit. We refer to the accumulator of this atomic MAC unit as the "inner" accumulator and denote its bit width as $P_I$. Conversely, we refer to the accumulator of the resulting partial sums as the "outer" accumulator and denote its bit width as $P_O$. Given that a $K$-dimensional dot product is executed in tiles of size $T$, where each tile is constrained to a $P_I$-bit accumulator, we can calculate the minimum bit width required to guarantee overflow avoidance for the outer accumulator as:

$$P_O = \lceil P_I + \log_2(K) - \log_2(T) \rceil \tag{21}$$

While we are the first to our knowledge to target multi-stage accumulation while guaranteeing overflow avoidance, accumulating in multiple stages is not new. Quantized inference libraries such as FBGEMM (Khudia et al., 2018a), XNNPACK (Dukahn & Barchard, 2021), and Ryzen AI (AMD, 2024) have employed multi-staged accumulation to exploit a 16-bit inner accumulator (*i.e.*, $P_I = 16$) to realize performance benefits, albeit without any theoretical guarantees of overflow avoidance. For example, Khudia et al. (2018b) use FBGEMM to realize nearly a $2\times$ throughput uplift on compute-bound workloads by accumulating at 16 bits in tiles of 64 elements rather than accumulating at 32 bits. Currently, these libraries typically disable this optimization if overflow is observed too often during testing. However, AXE provides a mechanism to simultaneously quantize and constrain a pre-trained model for low-precision multi-staged accumulation while guaranteeing overflow avoidance, enabling co-design for this optimization for the first time. As shown in Section 4, this generalization is critical in maintaining the quality of billion-parameter large language models, which often have dot products containing more than ten thousand elements.

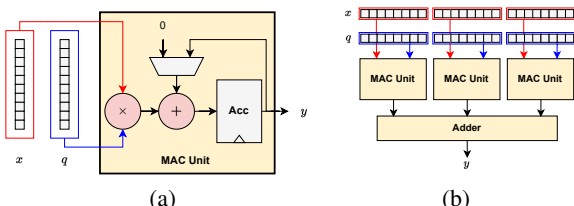

(a)  (b)

Figure 2: We visualize (a) an atomic MAC unit and (b) parallelized multi-staged accumulation.

## 4 EXPERIMENTS

**Models & Datasets.** We conduct experiments on GPT2 (Radford et al., 2019), OPT (Zhang et al., 2022a), and Pythia (Biderman et al., 2023) models and calibrate all quantized models using Wiki-Text2 (Merity et al., 2016). When focusing on our scaling analysis, we use the LM Evaluation Harness benchmarking suite (Gao et al., 2023) to evaluate 6 reasoning tasks: ARC-easy and ARC-challenge (Clark et al., 2018), HellaSwag (Zellers et al., 2019), LAMBADA (Radford et al., 2019), PIQA (Bisk et al., 2020), and Winogrande (Sakaguchi et al., 2021).

**Quantization Design Space.** We constrain our quantization design space to uniform-precision models such that every hidden layer has the same weight, activation, and accumulator bit width, respectively denoted as $M$, $N$, and $P$. We consider 3- to 8-bit integers for both weights and activations, unlike (Frantar et al., 2022) and (Zhang et al., 2023), which focused on weight-only quantization. Rather than evaluating each combination of $M$ and $N$, we restrict ourselves to configurations where $N \geq M$ to reduce the cost of experimentation as such configurations tend to dominate the Pareto frontiers (Colbert et al., 2024). We implement our methods in PyTorch (Paszke et al., 2019) using the Brevitas quantization library (Pappalardo, 2023). We include all hyperparameter details in Appendix C. All models are quantized using a single AMD MI210 GPU with 64 GB of memory.

### 4.1 OPTIMIZING FOR ACCUMULATOR CONSTRAINTS

We first consider the scenario in which QNNs are optimized for accumulator-constrained processors. Unlike prior work, we focus our analysis on the PTQ setting. As discussed in Section 2.3, one could heuristically manipulate $M$ and $N$ according to the data type bound derived by Colbert et al. (2023); however, the quantization design space ultimately limits the minimum attainable accumulator bit width. To the best of our knowledge, Euclidean projection-based initialization (EP-init) serves as the best alternative to this bit width manipulation approach in the PTQ setting (see Section 2.3). Therefore, we use EP-init and naïve bit width manipulation as our baselines.

In Figure 1, we use Pareto frontiers to visually characterize the trade-off between accumulator bit width $P$ and model quality for both GPFQ and OPTQ, respectively. We assume a monolithic accumulator in these experiments (*i.e.*, $P = P_I = P_O$). For each model and each PTQ algorithm, the Pareto frontier shows the lowest observed perplexity for a target $P$ when varying $M$ and $N$ within our design space, with the perplexity of the 32-bit floating-point model provided for reference. Recall that accumulator-aware quantization requires both the weights and activations to be quantized

Table 1: We report the WikiText2 perplexity results when evaluating AXE on Pythia models quantized to W4A8 for 16-bit accumulation in tiles of 128 elements (denoted 128×16b). We compare against the unconstrained baseline (denoted Base). We use our functionally equivalent memory-efficient GPFQ formulation to scale to larger models (see Appendix B).

|  |  | **70M** | **160M** | **410M** | **1.0B** | **1.4B** | **2.8B** | **6.9B** |
|---|---|---|---|---|---|---|---|---|
|  | **Float** | 45.2 | 26.7 | 15.9 | 13.2 | 11.8 | 10.2 | 9.2 |
| **GPFQ***| Base | 61.7 | 40.1 | 23.0 | 14.7 | 15.7 | 13.3 | 14.2 |
|  | 128×16b | 81.9 | 47.1 | 25.9 | 15.4 | 16.8 | 14.3 | 15.2 |
| **OPTQ** | Base | 65.4 | 46.6 | 28.9 | 14.7 | 15.7 | 17.3 | 13.5 |
|  | 128×16b | 201.4 | 131.8 | 60.7 | 16.2 | 18.6 | 16.6 | 16.2 |

(see Section 3.2); therefore, this is not a direct comparison against the original GPFQ and OPTQ proposals, which only quantized weights. We provide a detailed breakdown of each Pareto frontier in Appendix F, where we report the perplexity of each Pareto-dominant model, their weight and activation bit widths, and resulting unstructured weight sparsity. We observe similar trends as reported in Colbert et al. (2024); the Pareto-optimal activation bit width $N$ decreases as $P$ is reduced, and the unstructured weight sparsity conversely increases. This suggests that our accumulator-aware boundary constraints obey similar mechanics as the $\ell_1$-norm constraints of QAT methods, as our theoretical justification predicts (see Section 3.3).

## 4.2 LOW-PRECISION ACCUMULATION FOR LARGE LANGUAGE MODELS

As discussed in Colbert et al. (2024), the $\ell_1$-norm of an unconstrained weight vector inherently grows as its dimensionality increases. This suggests that accumulator-aware quantization scales well to strictly deeper neural architectures since the constraints tighten with width rather than depth; experimental results on the ResNet family support this hypothesis (Colbert et al., 2024). However, this also suggests that accumulator-aware quantization scales poorly in neural network families that grow in width, as is the case in transformer architectures (Zhang et al., 2022a; Biderman et al., 2023). Thus, to scale our accumulator-aware PTQ framework to billion-parameter language models, we turn to our multi-stage accumulation variant of AXE (see Section 3.3). Here, one assumes the partial sums of a dot product are concurrently computed in fixed-length tiles of size $T$. Our goal in this setting is to minimize perplexity for a target inner accumulator bit width $P_I$ that is assumed to be universal across all tiles. Thus, our accumulator width is constant even as models grow wider.

Rather than exploring the full quantization design space, we focus on 4-bit weights and 8-bit activations (W4A8) to maximize utility across platforms with a reasonable number of experiments, as prior studies have suggested this configuration is generally useful (Dettmers & Zettlemoyer, 2023; Li et al., 2024). We evaluate AXE on top of both GPFQ and OPTQ using tiles of 128 elements under 16-bit accumulator constraints (note that $P_I^* = 20$ when $T = 128$ for W4A8 via Eq. 3). Again, prior work has established 128 to be a generally useful tiling size; AVX-512 ISA supports $T = 32$ elements (Khudia et al., 2018a), Ryzen AI NPUs support $T = 64$ elements (AMD, 2024), and many works allocate scaling factors in groups of 128 elements (Lin et al., 2023; Liu et al., 2024). Finally, we find that the peak memory utilization of GPFQ limits its evaluation on billion-parameter LLMs. Thus, we introduce a functionality equivalent memory-efficient reformulation to enable the algorithm to scale to larger models (see Appendix B). We report our results in Tables 1 and 2.

We first focus our scaling law analysis on the Pythia model suite, which was specifically designed to facilitate such a study (Biderman et al., 2023). From our results in Table 1, we observe that, as model size increases, the quality of the accumulator-constrained models approaches that of the unconstrained baselines as expected, with AXE preserving 92.4% of the baseline GPFQ perplexity and 82.8% of the baseline OPTQ perplexity of Pythia-6.9B compared to 74.8% and 32.3% for Pythia-70M, respectively. From our results in Table 2, we again observe that, as model size increases, the gap is reduced between the zero-shot reasoning capabilities of the constrained models and their unconstrained baselines; AXE preserves 98% of the baseline GPFQ accuracy and 96% of the baseline OPTQ accuracy for Pythia-6.9B compared to 86% and 76% for Pythia-70M, respectively. Under the A2Q scaling hypothesis, this suggests the narrowing accuracy gap is in part because model capacity is growing without tightening the constraints since $T$ is held constant even as $K$ increases. In Ap-

Table 2: We report the geometric mean calculated over 6 zero-shot reasoning tasks when evaluating AXE on Pythia models quantized to W4A8 for 16-bit accumulation in tiles of 128 elements (denoted $128{\times}16b$). We compare against the unconstrained baseline (denoted Base). We again use our functionally equivalent memory-efficient GPFQ formulation to scale to larger models (see Appendix B).

|  |  | **70M** | **160M** | **410M** | **1.0B** | **1.4B** | **2.8B** | **6.9B** |
|---|---|---|---|---|---|---|---|---|
|  | **Float** | 32.9 | 40.1 | 43.6 | 47.3 | 50.2 | 53.9 | 56.1 |
| **GPFQ***  | Base | 26.4 | 33.7 | 34.0 | 42.4 | 39.5 | 44.1 | 40.0 |
|  | $128{\times}16b$ | 22.8 | 32.5 | 31.5 | 41.9 | 37.7 | 42.9 | 39.3 |
| **OPTQ** | Base | 29.0 | 36.6 | 38.4 | 43.9 | 44.7 | 46.5 | 47.3 |
|  | $128{\times}16b$ | 22.2 | 32.6 | 29.3 | 40.0 | 43.1 | 45.8 | 45.6 |

pendix C.2, we provide an ablation study targeting a monolithic 16-bit accumulator (*i.e.*, $P_O = 16$). There, we show the gap conversely increases as $K$ increases, confirming that fixing $P_I$ via multi-stage accumulation improves scaling.

## 5 DISCUSSION AND CONCLUSIONS

As neural networks continue to increase in size, and their weights and activations are increasingly being represented with fewer bits, we anticipate the accumulator to play a larger role in hardware-software co-design (see Section 2.2). While prior work on accumulator-aware quantization has been limited to the QAT setting (see Section 2.3), ours marks the first to extend accumulator-awareness to the PTQ setting. To do so, we introduce AXE—a practical low-overhead framework of accumulator-aware extensions designed to endow overflow avoidance guarantees to any layer-wise PTQ algorithm that greedily assign bits element-by-element. We demonstrate the flexibility of AXE by presenting accumulator-aware variants of GPFQ and OPTQ with principled overflow avoidance guarantees. Furthermore, unlike prior accumulator-aware quantization methods, which assume a monolithic accumulator, we generalize AXE to support multi-stage accumulation for the first time.

Our experiments in Section 4.1 show that AXE significantly improves the trade-off between accumulator bit width and model accuracy when compared to existing baselines. As has been shown before in the QAT setting (Colbert et al., 2023; 2024), exposing control over the accumulator bit width allows one to reduce $P$ further than what is attainable via naïve bit width manipulations while also maintaining model accuracy. Moreover, we observe that AXE universally yields marked improvement over EP-init across both models and datasets, establishing a new state-of-the-art for accumulator-aware quantization in the PTQ setting. Although EP-init and AXE are both derived from the same convex optimization problem (see Eq. 14), EP-init is a vector projection that is applied after quantization and relies on the round-to-zero rounding function to ensure the $\ell_1$-norm constraints are respected. Previous reports had suspected EP-init is limited by this reliance on round-to-zero (Colbert et al., 2023; 2024); we provide an ablation study in Appendix C.2 that supports this hypothesis but also suggests error correction is critical. For GPFQ, we observe error correction to be more important than round-to-nearest, but we observe the opposite for OPTQ, although a more exhaustive analysis in future work may uncover more insights.

Our experiments in Section 4.2 show that our generalized multi-stage accumulation enables accumulator-aware weight quantization for billion-parameter LLMs. We observe that the gap between the constrained and unconstrained quantized models shrinks as model size increases, preserving both perplexity and zero-shot reasoning. However, we also observe that the gap between the quantized models and their 32-bit floating-point counterparts begins to increase with model size. This is consistent with the findings of Li et al. (2024), who conclude that while larger models tend to have a higher tolerance for weight quantization, they also tend to have a lower tolerance for activation quantization. Thus, there exists two diametrically opposing trends in superposition. For the Pythia model suite, we observe Pythia-1B to be the equilibrium point where the costs of weight and activation quantization are balanced. While it is orthogonal to the scope of this study, we expect the emerging rotation-based quantization schemes (*e.g.*, QuaRot (Ashkboos et al., 2024) or Spin-Quant (Liu et al., 2024)) to impact this equilibrium point and reduce the gap between quantized models and their 32-bit floating-point counterparts. We leave such investigations for future work.

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

# A  PSUEDO-CODE FOR ACCUMULATOR-AWARE VARIANTS OF GPFQ AND OPTQ

We present the pseudo-code for our accumulator-aware variants of GPFQ (Lybrand & Saab, 2021) and OPTQ (Frantar et al., 2022) in Algorithms 1 and 2, respectively, where we define $\Psi_{\boldsymbol{a},\boldsymbol{b}}(\boldsymbol{v})$ to denote the clipping function applied elementwise so that $(\Psi_{\boldsymbol{a},\boldsymbol{b}}(\boldsymbol{v}))_j = \Psi_{a_j,b_j}(v_j)$. We direct the reader to Section 3 for theoretical justification for these algorithms.

---

**Algorithm 1 Accumulator-Aware GPFQ.** Our accumulator-aware GPFQ variant (Lybrand & Saab, 2021) quantizes $\boldsymbol{W}$ to $M$ bits given input activations $\boldsymbol{X}$ and their $N$-bit quantized counterparts $\tilde{\boldsymbol{X}}$. Note that $\boldsymbol{W}_i, \boldsymbol{V}_i \in \mathbb{R}^C$, $\boldsymbol{Q}_i \in \mathcal{A}_M^C$, $\boldsymbol{X}_i \in \mathbb{R}^D$, and $\tilde{\boldsymbol{X}}_i \in \mathcal{A}_N^D$, all interpreted as row vectors.

---

**Require:** $\boldsymbol{W} \in \mathbb{R}^{K \times C}$, $\boldsymbol{X} \in \mathbb{R}^{K \times D}$, $\tilde{\boldsymbol{X}} \in \mathcal{A}_N^{K \times D}$

1: $\boldsymbol{Q} \leftarrow 0 \in \mathcal{A}_M^{K \times C}$.                                          // *Quantized output*
2: $\boldsymbol{U} \leftarrow 0 \in \mathbb{R}^{D \times C}$                                           // *Per-sample quantization error*
3: $\boldsymbol{a} \leftarrow A \in \mathbb{R}^C$, $\boldsymbol{b} \leftarrow B \in \mathbb{R}^C$       // *Initialize running sums*
4: $\lambda \leftarrow \text{deriveThreshold}(\boldsymbol{W})$                                         // *Derive per-channel Lagrangian thresholds*
5: **for** $i = 1, ..., K$ **do**
6:     $\boldsymbol{V}_i \leftarrow \boldsymbol{W}_i \frac{\langle \tilde{\boldsymbol{X}}_i, \boldsymbol{X}_i \rangle}{\|\tilde{\boldsymbol{X}}_i\|_2^2} + \frac{\tilde{\boldsymbol{X}}_i \boldsymbol{U}}{\|\tilde{\boldsymbol{X}}_i\|_2^2}$     // *Adjust for quantization error*
7:     $\boldsymbol{V}_i \leftarrow \Psi_{\boldsymbol{a},\boldsymbol{b}} \circ \Pi_\lambda(\boldsymbol{V}_i)$       // *Accumulator-aware projection & clipping*
8:     $\boldsymbol{Q}_i \leftarrow \mathcal{Q}(\boldsymbol{V}_i)$                                       // *Quantize weight*
9:     $\boldsymbol{a} \leftarrow \boldsymbol{a} - \boldsymbol{Q}_i \odot \mathbb{1}_{\boldsymbol{Q}_i \geq 0}$     // *Update positive range*
10:    $\boldsymbol{b} \leftarrow \boldsymbol{b} - \boldsymbol{Q}_i \odot \mathbb{1}_{\boldsymbol{Q}_i \leq 0}$     // *Update negative range*
11:    $\boldsymbol{U} \leftarrow \boldsymbol{U} + \boldsymbol{X}_i^T \boldsymbol{W}_i - \tilde{\boldsymbol{X}}_i^T \boldsymbol{Q}_i$     // *Update quantization error*
12: **end for**
13: **return** $\boldsymbol{Q}$

---

**Algorithm 2 Accumulator-Aware OPTQ.** Our accumulator-aware OPTQ variant (Frantar et al., 2022) quantizes $\boldsymbol{W}$ to $M$ bits given $\boldsymbol{H}^{-1} = \text{Cholesky}((2\tilde{\boldsymbol{X}}\tilde{\boldsymbol{X}}^T + \eta\boldsymbol{I})^{-1})$, where $\eta$ is a small dampening factor to avoid numerical issues. Following Frantar et al. (2022), we set $\eta$ to be $1\%$ of the average diagonal value. Note that $\boldsymbol{W}_i, \boldsymbol{V}_i \in \mathbb{R}^C$ and $\boldsymbol{Q}_i \in \mathcal{A}_M^C$, all interpreted as row vectors.

---

**Require:** $\boldsymbol{W} \in \mathbb{R}^{K \times C}$, $\boldsymbol{H}^{-1} \in \mathbb{R}^{K \times K}$

1: $\boldsymbol{Q} \leftarrow 0 \in \mathcal{A}_M^{K \times C}$                                          // *Quantized output*
2: $\boldsymbol{E} \leftarrow 0 \in \mathbb{R}^C$                                                       // *Per-channel quantization errors*
3: $\boldsymbol{a} \leftarrow A \in \mathbb{R}^C$, $\boldsymbol{b} \leftarrow B \in \mathbb{R}^C$       // *Initialize running sums*
4: $\lambda \leftarrow \text{deriveThreshold}(\boldsymbol{W})$                                         // *Derive per-channel Lagrangian thresholds*
5: **for** $i = 1, ..., K$ **do**
6:     $\boldsymbol{V}_i \leftarrow \Psi_{\boldsymbol{a},\boldsymbol{b}} \circ \Pi_\lambda(\boldsymbol{W}_i)$        // *Accumulator-aware projection & clipping*
7:     $\boldsymbol{Q}_i \leftarrow \mathcal{Q}(\boldsymbol{V}_i)$                                       // *Quantize processed weight*
8:     $\boldsymbol{E} \leftarrow (\boldsymbol{W}_i - \boldsymbol{Q}_i)/\boldsymbol{H}_{i,i}^{-1}$       // *Calculate quantization error*
9:     $\boldsymbol{W}_{i:K} \leftarrow \boldsymbol{W}_{i:K} - \boldsymbol{E} \cdot \boldsymbol{H}_{i,i:K}^{-1}$     // *Update weights*
10:    $\boldsymbol{a} \leftarrow \boldsymbol{a} - \boldsymbol{Q}_i \odot \mathbb{1}_{\boldsymbol{Q}_i \geq 0}$     // *Update positive range*
11:    $\boldsymbol{b} \leftarrow \boldsymbol{b} - \boldsymbol{Q}_i \odot \mathbb{1}_{\boldsymbol{Q}_i \leq 0}$     // *Update negative range*
12: **end for**
13: **return** $\boldsymbol{Q}$

---

# B  MEMORY-EFFICIENT GPFQ

As discussed in Section 3.2, GPFQ approaches the standard quantization problem by traversing the neural network graph to sequentially quantize each element in each layer while iteratively correcting

for quantization error. The derived iteration rule is formalized by Eqs. 10 and 11. In this standard formulation, the $i$-th quantized weight $q_i$ depends on the inner product

$$\langle \tilde{\boldsymbol{X}}_i^{(l)}, \boldsymbol{u}_{i-1}^{(l)} + w_i^{(l)} \boldsymbol{X}_i^{(l)} \rangle$$

where $\boldsymbol{X}_i^{(l)}, \tilde{\boldsymbol{X}}_i^{(l)} \in \mathbb{R}^D$ are samples for the $i$-th neuron of the inputs to layer $l$, and $\boldsymbol{u}_{i-1}^{(l)} \in \mathbb{R}^D$ is the running error from quantizing the first $i - 1$ weights. Thus, at layer $l$, GPFQ requires collecting and storing $2D$ samples for the $K_l$ input neurons, and updating the running quantization error for each sample for the $C_l$ output neurons. This implies poor scaling to larger models and larger calibration sets as the memory requirements are $O(D \times (2K_l + C_l))$. Indeed, assuming 128 samples with a sequence length of 2048 at 32-bit precision, Pythia-6.9B (Biderman et al., 2023) requires a peak memory usage of roughly 30 GB at the first FFN layer excluding pre-trained weights. We set out to reduce this overhead.

We start with the observation that OPTQ is far more memory efficient. OPTQ uses the Hessian proxy $2\boldsymbol{X}\boldsymbol{X}^T$, which can be efficiently computed one sample at a time and stored as a $K_l \times K_l$ square matrix, an $O(K_l \times K_l)$ memory requirement that is $36\times$ less than GPFQ for Pythia-6.9B. Thus, we reformulate GPFQ to use square matrices via mathematical manipulation of singular value decompositions. We present the following theorem:

**Theorem B.1.** *Let* $\boldsymbol{H} = \left(\tilde{\boldsymbol{X}}\tilde{\boldsymbol{X}}^T\right)^{1/2}$ *and* $\boldsymbol{G} = \boldsymbol{X}\tilde{\boldsymbol{X}}^T$. *For pre-trained weights* $\boldsymbol{W} \in \mathbb{R}^{K \times C}$, *quantization alphabet* $\mathcal{A}$, *and GPFQ function of the form of Algorithm 1, it follows that:*

$$GPFQ(\boldsymbol{W}, \boldsymbol{X}, \tilde{\boldsymbol{X}}, \mathcal{A}) = GPFQ(\boldsymbol{W}, \boldsymbol{G}\boldsymbol{H}^{-1}, \boldsymbol{H}, \mathcal{A}) \tag{22}$$

*Proof.* According to the iteration steps in Algorithm 1, it suffices to show that the argument of quantizer $\mathcal{Q}$ is unchanged after substituting $\boldsymbol{X}_i, \tilde{\boldsymbol{X}}_i$ with $(\boldsymbol{G}\boldsymbol{H}^{-1})_i$ and $\boldsymbol{H}_i$ respectively. Specifically, at the $i$-th iteration of $GPFQ(\boldsymbol{W}, \boldsymbol{G}\boldsymbol{H}^{-1}, \boldsymbol{H}, \mathcal{A})$, we have

$$\boldsymbol{V}_i \leftarrow \boldsymbol{W}_i \frac{\langle \boldsymbol{H}_i, (\boldsymbol{G}\boldsymbol{H}^{-1})_i \rangle}{\|\boldsymbol{H}_i\|_2^2} + \frac{\boldsymbol{H}_i \boldsymbol{U}_{i-1}}{\|\boldsymbol{H}_i\|_2^2} \tag{23}$$

where the quantization error is given by

$$\boldsymbol{U}_{i-1} = \sum_{j=1}^{i-1} (\boldsymbol{G}\boldsymbol{H}^{-1})_j^T \boldsymbol{W}_j - \boldsymbol{H}_j^T \boldsymbol{Q}_j. \tag{24}$$

Let $\boldsymbol{e}_i \in \mathbb{R}^K$ denote the vector with a 1 in the $i$-th coordinate and 0's elsewhere. It follows from $\boldsymbol{H} = \left(\tilde{\boldsymbol{X}}\tilde{\boldsymbol{X}}^T\right)^{1/2}$ and $\boldsymbol{G} = \boldsymbol{X}\tilde{\boldsymbol{X}}^T$ that

$$\|\boldsymbol{H}_i\|_2^2 = \|\boldsymbol{e}_i^T \boldsymbol{H}\|_2^2 = \boldsymbol{e}_i^T \boldsymbol{H}^2 \boldsymbol{e}_i = \boldsymbol{e}_i^T \tilde{\boldsymbol{X}}\tilde{\boldsymbol{X}}^T \boldsymbol{e}_i = \|\tilde{\boldsymbol{X}}_i\|_2^2,$$

$$\boldsymbol{H}_i (\boldsymbol{G}\boldsymbol{H}^{-1})_j^T = \boldsymbol{e}_i^T \boldsymbol{H} (\boldsymbol{e}_j^T \boldsymbol{G}\boldsymbol{H}^{-1})^T = \boldsymbol{e}_i^T \boldsymbol{G}^T \boldsymbol{e}_j = \boldsymbol{e}_i^T \tilde{\boldsymbol{X}}\boldsymbol{X}^T \boldsymbol{e}_j = \tilde{\boldsymbol{X}}_i \boldsymbol{X}_j^T,$$

and

$$\boldsymbol{H}_i \boldsymbol{H}_j^T = \boldsymbol{e}_i^T \boldsymbol{H} (\boldsymbol{e}_j^T \boldsymbol{H})^T = \boldsymbol{e}_i^T \boldsymbol{H}^2 \boldsymbol{e}_j = \boldsymbol{e}_i^T \tilde{\boldsymbol{X}}\tilde{\boldsymbol{X}}^T \boldsymbol{e}_j = \tilde{\boldsymbol{X}}_i \tilde{\boldsymbol{X}}_j^T.$$

Plugging above identities into equation 23 and equation 24, we obtain

$$\boldsymbol{V}_i \leftarrow \boldsymbol{W}_i \frac{\langle \tilde{\boldsymbol{X}}_i, \boldsymbol{X}_i \rangle}{\|\tilde{\boldsymbol{X}}_i\|_2^2} + \frac{\tilde{\boldsymbol{X}}_i \hat{\boldsymbol{U}}_{i-1}}{\|\tilde{\boldsymbol{X}}_i\|_2^2} \tag{25}$$

with $\hat{\boldsymbol{U}}_{i-1} = \sum_{j=1}^{i-1} \boldsymbol{X}_j^T \boldsymbol{W}_j - \tilde{\boldsymbol{X}}_j^T \boldsymbol{Q}_j$. Since $\boldsymbol{V}_i$ in equation 25 is identical with the $i$-th quantization argument in $GPFQ(\boldsymbol{W}, \boldsymbol{X}, \tilde{\boldsymbol{X}}, \mathcal{A})$, both algorithms derive the same quantized weights $\boldsymbol{Q}_i = \mathcal{Q}(\boldsymbol{V}_i)$. This completes the proof. □

At layer $l$, this memory-efficient GPFQ formulation requires collecting and storing $G$, $H$, and $U$, which are each $K_l \times K_l$ matrices, reducing to an $O(K_l \times K_l)$ memory requirement that is $12\times$ less than the standard GPFQ formulation for Pythia-6.9B. We leverage this functionally equivalent formulation for our LLM evaluations in Section 4.2.

## C EXPERIMENTAL DETAILS & ABLATIONS

### C.1 HYPERPARAMETERS & QUANTIZATION SCHEMES

Below, we provide a detailed description of the quantization schemes and the specific hyperparameters used in our experiments. As discussed in Section 4, we consider pre-trained autoregressive language models that are respectively made publicly available via the HuggingFace (Wolf et al., 2020) libraries. All models are quantized via the Brevitas (Pappalardo, 2023) quantization library using a single AMD MI210 GPU with 64 GB of memory.

We leverage the unmodified implementations of the various LLMs discussed in Section 4 as provided by HuggingFace (Wolf et al., 2020), as well as their pre-trained floating-point checkpoints and datasets (Lhoest et al., 2021). We use drop-in replacements for all linear layers in the networks except the embedding layer or final prediction head, leaving them at 32-bit floating-point. As is common practice (Frantar et al., 2022), we build our calibration set using 128 samples randomly selected from the WikiText2 dataset (Merity et al., 2016) without replacement using a fixed sequence length of 2048 tokens for all models except GPT2 (Radford et al., 2019), which is restricted to a maximum sequence length of 1024 by the library.

**Quantization Scheme.** As discussed in Section 2, we adopt the standard uniform integer quantizer parameterized by scaling factor $s$ and zero-point $z$. We quantize activations asymmetrically, tuning $z$ to the lowest 99-th percentile based on the calibration data. While AXE is not reliant on symmetric weight quantization, we eliminate zero-points in all weight quantizers such that $z = 0$, as is common practice so as to avoid computational overhead of cross-terms (Nagel et al., 2021; Zhang et al., 2022b). Throughout our experiments, we adopt 32-bit floating-point scaling factors that take the form of Eq. 26, where $\max(\boldsymbol{w})$ is calculated *per-channel* for the weights and *per-tensor* for the activations quantized to $b$ bits.

$$s = \frac{\max(\boldsymbol{w})}{2^{b-1} - 1} \tag{26}$$

**Quantization Process.** To quantize our models, we first load the pre-trained checkpoint and merge batch normalization layers if they exist, then we apply SmoothQuant (Xiao et al., 2023) before calibrating the scaling factors and zero-points. We then apply either GPFQ (Lybrand & Saab, 2021) or OPTQ (Frantar et al., 2022) (with or without AXE) before finally applying bias correction (Nagel et al., 2019). When sequentially quantizing weights element-by-element, we do so in descending order according to the diagonal value of the Hessian proxy ($2\boldsymbol{X}\boldsymbol{X}^T$ by our notation in Section 2), which was originally implemented in IST-DASLab (2022) and reported to yield superior results in Lin et al. (2023); Chee et al. (2024). When evaluating EP-init in the PTQ setting, we do so after OPTQ or GPFQ but before bias correction. Because bias correction does not adjust weight values, this allows us to at least perform some form of error correction with EP-init while still ensuring guaranteed overflow avoidance.

### C.2 ABLATION STUDIES

**Impact of error correction and choice of rounding function.** Previous reports had suspected EP-init is limited by its reliance on the round-to-zero (RTZ) rounding function (Colbert et al., 2023; 2024), which has been shown to be a poor choice (Nagel et al., 2020) AXE removes this reliance and also enables greedy error correction. We design an ablation study to isolate the impact of RTZ and error correction. We quantize OPT-125M (Zhang et al., 2022a) and Pythia-160M (Biderman et al., 2023) to 4-bit weights and 8-bit activations while targeting 20-bit accumulation since our Pareto front shows this configuration to be both reasonable and challenging. We evaluate AXE with round-to-zero (AXE-RTZ) and AXE with round-to-nearest (AXE-RTN). We report the results in Table 3. We interpret the gap between EP-init and AXE-RTZ as the benefit of error correction, and the gap between AXE-RTZ and AXE-RTN as the benefit of rounding function. We observe that error correction has a greater impact than rounding function selection for GPFQ, but we observe the opposite for OPTQ. Finally, we evaluate AXE with our hard constraint only (AXE-HCO) to isolate the impact of our soft constraint, which is not necessary for guaranteeing overflow avoidance. We interpret the gap between AXE-RTN and AXE-HCO as the impact of our soft constraint, which consistently provides improved or maintained performance.

Table 3: We evaluate round-to-nearest (RTN) and round-to-zero (RTZ) within our AXE framework to directly compare against EP-init. We also evaluate AXE with our hard constraint only (HCO) to isolate the impact of our soft constraint. All models are quantized to W4A8 while targeting a 20-bit monolitic accumulator (*i.e.*, $P_O = 20$).

| Algorithm | Model | EP-init | AXE-RTZ | AXE-RTN | AXE-HCO |
|---|---|---|---|---|---|
| GPFQ | OPT-125M | 8828.3 | 165.2 | 31.9 | 31.9 |
| | Pythia-160M | 2500.2 | 211.0 | 43.0 | 49.2 |
| OPTQ | OPT-125M | 998.6 | 539.3 | 37.1 | 70.0 |
| | Pythia-160M | 4524.4 | 1798.7 | 84.9 | 194.8 |

**Multi-stage vs. monolithic accumulation.** In Section 4.2, we analyze how our accumulator constraints scale to increasingly large language models within the Pythia model suite (Biderman et al., 2023). There, we discuss our observation that, as model size increases, the quality of the accumulator-constrained models approaches that of the unconstrained baselines for both GPFQ and OPTQ. This suggests the narrowing gap in perplexity is in part because model capacity is growing without tightening the constraints. To verify this, we perform an ablation study targeting a monolithic 16-bit accumulator (*i.e.*, $P_I = P_O = 16$). We quantize all Pythia models up to Pythia-1B using either OPTQ or GPFQ, and report the results in Table 4. Not only do we observe significant instability, we also observe a $7.4\times$ regression in perplexity between Pythia-70M and Pythia-1B, confirming that fixing $P_I$ improves scaling as models grow wider.

Table 4: We evaluate AXE using Pythia models quantized to W4A8 when targeting a monolithic 16-bit accumulator (*i.e.*, $P_O = 16$). Note that this is in direct contrast with Table 1, which targets multi-stage accumulation (*i.e.*, $P_I = 16$).

| Algorithm | 70M | 160M | 410M | 1B |
|---|---|---|---|---|
| GPFQ | 4397 | 7135 | 10496 | 32601 |
| OPTQ | 2438 | 4439 | 9759 | 34387 |

# D    ADDITIONAL EXPERIMENTS WITH LLAMA3

Our intention with focusing on Pythia in Section 4.2 was to investigate scaling, for which the Pythia model family was specifically designed (Biderman et al., 2023). However, to demonstrate generalization to another model family, we provide additional results with Llama3 (Dubey et al., 2024) evaluated on WikiText2 (Merity et al., 2016).

To the best of our knowledge, only datatype manipulation and EP-init (Colbert et al., 2024) serve as alternatives to AXE for accumulator-aware quantization in the PTQ setting. As shown in Figure 1, AXE is the Pareto-dominant algorithm. Furthermore, as discussed in Section 4.2, we observe that multi-stage accumulation is critical to ensure accumulator-aware quantization scales to increasingly large language models (also see Appendix C.2 for ablations). Therefore, as EP-init does not support multi-stage accumulation, the only existing alternative accumulator-aware PTQ mechanism for billion-parameter LLMs is datatype manipulation. Note that, via Eq. 3, W4A4 guarantees overflow avoidance for 16-bit accumulation in tiles of 128 elements. Therefore, we compare AXE to datatype manipulation when constraining a model to target 16-bit accumulation in tiles of 128 elements (*i.e.*, $128 \times 16b$). We provide our perplexity results in Table 5 along with the 32-bit accumulator baselines (*i.e.*, $128 \times 32b$) as well as the original 32-bit floating-point perplexities.

Note that AXE enables low-precision accumulation for Llama3 with minimal degradation from the unconstrained baselines. As discussed in Section 3.2, AXE has the desired feature of being functionally equivalent to the underlying algorithm (*e.g.*, OPTQ or GPFQ) when the accumulator is large enough (*e.g.*, 32 bits). Thus, one should expect these benefits to manifest most when targeting low-precision accumulators (*e.g.*, 16 bits) but not high-precision accumulators (*e.g.*, 32 bits), as observed in Table 5. Furthermore, the gap between the constrained and unconstrained baseline decreases as the model size increases. This result supports our scaling hypothesis in Section 4.2 as well as our results with the Pythia model family.

Table 5: We report the WikiText2 perplexity when evaluating Llama3 models quantized for 16-bit accumulation in tiles of 128 elements (denoted "128 × 16b"). We compare AXE against datatype manipulation (denoted "Base"), which serves as the only alternative for billion-parameter models. Note the 128 × 32b baseline is W4A8 while the 128 × 16b baseline is W4A4.

|  |  | **3.2-1B** | **3.2-3B** | **3.1-8B** |
|---|---|---|---|---|
|  | **Float16** | 11.8 | 9.1 | 6.5 |
| **OPTQ** | Base | 14.5 | 10.2 | 7.5 |
| (128×32b) | AXE | 14.4 | 10.2 | 7.5 |
| **OPTQ** | Base | inf | inf | inf |
| (128×16b) | AXE | 14.9 | 10.4 | 7.6 |

To collect these results, we use the same quantization scheme discussed in Appendix C aside from using *per-token* dynamic activation scaling rather than *per-tensor*, which improves model quality without impacting our accumulator constraint guarantees. We use the same quantization process described in Appendix C, but remove bias correction, which seems to have minimal impact on these models. We perform a light grid search over SmoothQuant's $\alpha$ parameter and find $\alpha = 0.4$ to generally perform the best on average for these models.

## E  ZERO-SHOT REASONING DETAILS

We provide the detailed zero-shot reasoning results presented in Section 4.2 for GPFQ and OPTQ. We present the results for the Pythia model suite in Table 6. We quantize all models to W4A8 and use AXE to constrain quantized models for 16-bit multi-stage accumulation in tiles of size 128 elements (denoted 128×16b). We compare against the unconstrained baselines (denoted Base). We report the geometric average calculated over 6 reasoning task evaluations: ARC-easy (ARC-E) and ARC-challenge (ARC-C) (Clark et al., 2018), HellaSwag (HS) (Zellers et al., 2019), LAMBADA (LA) (Radford et al., 2019), PIQA (Bisk et al., 2020), and Winogrande (Wino) (Sakaguchi et al., 2021). We use the LM Evaluation Harness benchmarking suite (Gao et al., 2023) for zero-shot reasoning without changing other default parameters. We use our functionally equivalent memory-efficient GPFQ formulation to scale to larger language models (see Appendix B).

## F  PARETO FRONTIER DETAILS

We provide the detailed Pareto frontiers visualized in Figure 1 for GPFQ and OPTQ. For each model, we report the perplexity, quantization configuration, and unstructured weight sparsity.

Table 6: We provide the details of the zero-shot reasoning tasks presented in Section 4.2.

| Model | Algorithm | Variant | PPL | Avg. | ARC-C | ARC-E | HS | LA | PIQA | Wino |
|---|---|---|---|---|---|---|---|---|---|---|
| **Pythia-70M** | **Float** | - | 45.2 | 32.9 | 17.5 | 37.5 | 26.7 | 22.7 | 59.8 | 52.9 |
| | **GPFQ\*** | Base | 61.7 | 26.4 | 16.1 | 35.6 | 26.8 | 7.3 | 58.1 | 51.7 |
| | | 128×16b | 81.9 | 22.8 | 17.8 | 34.2 | 26.4 | 2.9 | 57.5 | 51.9 |
| | **OPTQ** | Base | 65.4 | 29.0 | 17.5 | 36.0 | 26.5 | 11.8 | 58.4 | 52.2 |
| | | 128×16b | 201.4 | 22.2 | 18.6 | 34.1 | 26.6 | 2.4 | 55.9 | 51.9 |
| **Pythia-160M** | **Float** | - | 26.7 | 34.9 | 19.5 | 43.6 | 28.4 | 35.4 | 62.3 | 51.3 |
| | **GPFQ\*** | Base | 40.1 | 26.6 | 19.2 | 39.4 | 27.9 | 6.5 | 60.2 | 49.2 |
| | | 128×16b | 47.1 | 18.0 | 18.3 | 38.4 | 27.6 | 0.4 | 58.5 | 52.1 |
| | **OPTQ** | Base | 46.6 | 31.4 | 19.2 | 39.4 | 28.0 | 18.7 | 61.2 | 53.0 |
| | | 128×16b | 131.8 | 24.1 | 21.0 | 36.0 | 27.3 | 3.4 | 58.3 | 49.3 |
| **Pythia-410M** | **Float** | - | 15.9 | 43.6 | 21.4 | 51.9 | 33.7 | 51.6 | 66.7 | 53.4 |
| | **GPFQ\*** | Base | 23.0 | 34.0 | 20.1 | 46.0 | 31.8 | 15.7 | 63.7 | 52.6 |
| | | 128×16b | 25.9 | 31.5 | 18.9 | 41.0 | 30.3 | 12.5 | 62.1 | 53.4 |
| | **OPTQ** | Base | 28.9 | 38.4 | 20.7 | 47.5 | 32.2 | 29.7 | 64.2 | 52.7 |
| | | 128×16b | 60.7 | 29.3 | 20.7 | 40.7 | 30.2 | 8.0 | 60.6 | 51.5 |
| **Pythia-1.0B** | **Float** | - | 13.2 | 47.3 | 24.4 | 57.0 | 37.8 | 56.3 | 70.7 | 53.4 |
| | **GPFQ\*** | Base | 14.7 | 42.4 | 21.0 | 50.2 | 35.2 | 44.5 | 65.8 | 53.2 |
| | | 128×16b | 15.4 | 41.9 | 20.9 | 50.8 | 34.3 | 44.4 | 65.1 | 51.9 |
| | **OPTQ** | Base | 14.7 | 43.9 | 22.4 | 51.0 | 36.5 | 46.9 | 67.2 | 54.3 |
| | | 128×16b | 16.2 | 40.0 | 22.2 | 48.0 | 35.2 | 32.4 | 65.0 | 51.8 |
| **Pythia-1.4B** | **Float** | - | 11.8 | 50.2 | 26.1 | 60.5 | 40.4 | 61.7 | 70.8 | 57.5 |
| | **GPFQ\*** | Base | 15.7 | 39.5 | 23.3 | 53.5 | 36.4 | 22.4 | 66.5 | 55.9 |
| | | 128×16b | 16.8 | 37.7 | 22.4 | 52.6 | 35.8 | 18.4 | 65.7 | 56.3 |
| | **OPTQ** | Base | 15.7 | 44.7 | 23.7 | 55.2 | 38.5 | 42.6 | 67.2 | 55.6 |
| | | 128×16b | 18.6 | 43.1 | 23.1 | 53.5 | 37.1 | 39.4 | 65.9 | 53.9 |
| **Pythia-2.8B** | **Float** | - | 10.2 | 53.9 | 29.4 | 64.4 | 45.3 | 64.7 | 74.0 | 60.1 |
| | **GPFQ\*** | Base | 13.3 | 44.1 | 26.2 | 57.1 | 40.7 | 30.9 | 69.6 | 56.2 |
| | | 128×16b | 14.3 | 42.9 | 24.2 | 55.6 | 40.0 | 28.4 | 70.2 | 58.2 |
| | **OPTQ** | Base | 17.3 | 46.5 | 26.7 | 59.4 | 42.4 | 37.0 | 70.7 | 57.7 |
| | | 128×16b | 16.6 | 45.8 | 26.0 | 58.2 | 41.0 | 37.5 | 68.8 | 57.5 |
| **Pythia-6.9B** | **Float** | - | 9.2 | 56.1 | 31.5 | 67.3 | 48.1 | 67.1 | 75.2 | 60.7 |
| | **GPFQ\*** | Base | 14.2 | 40.0 | 28.5 | 56.5 | 40.9 | 15.4 | 70.2 | 57.4 |
| | | 128×16b | 15.2 | 39.3 | 27.2 | 55.8 | 39.9 | 15.1 | 69.4 | 57.9 |
| | **OPTQ** | Base | 13.5 | 47.3 | 29.1 | 62.8 | 45.4 | 32.9 | 70.7 | 57.9 |
| | | 128×16b | 16.2 | 45.6 | 29.9 | 59.7 | 44.0 | 27.0 | 70.7 | 59.8 |

Table 7: **GPFQ:** We provide the test perplexity (PPL) and quantization configuration of the Pareto-optimal models that form the frontiers visualized in Figure 1. Note that $M$ and $N$ respectively denote the weight and activation bit widths.

| Model | P | GPFQ | | | GPFQ+EP-init | | | GPFQ+AXE | | |
|---|---|---|---|---|---|---|---|---|---|---|
| | | PPL | $(M,N)$ | Sparsity | PPL | $(M,N)$ | Sparsity | PPL | $(M,N)$ | Sparsity |
| **OPT-125M** (Float: 27.7) | 16 | - | - | - | 9148.8 | (3,4) | 76.5 | **249.8** | **(3,6)** | **55.6** |
| | 17 | - | - | - | 7624.6 | (3,4) | 72.7 | **91.2** | **(4,6)** | **37.9** |
| | 18 | 11007.2 | (3,3) | 58.3 | 7471.2 | (3,5) | 75.5 | **41.8** | **(4,6)** | **27.8** |
| | 19 | 9567.6 | (3,4) | 54.5 | 1059.3 | (5,6) | 39.1 | **32.3** | **(4,7)** | **27.0** |
| | 20 | 874.4 | (3,5) | 50.5 | 86.1 | (5,6) | 29.8 | **29.3** | **(5,7)** | **15.7** |
| | 21 | 101.0 | (3,6) | 46.4 | 42.4 | (5,7) | 28.1 | **28.6** | **(5,8)** | **15.6** |
| | 22 | 40.5 | (4,6) | 26.3 | 30.4 | (6,7) | 16.0 | **28.1** | **(6,8)** | **9.6** |
| | 23 | 31.8 | (4,7) | 25.9 | 29.5 | (6,8) | 15.9 | **27.9** | **(6,8)** | **8.6** |
| | 24 | 29.0 | (5,7) | 14.7 | 28.2 | (7,8) | 9.5 | **27.8** | **(7,8)** | **5.4** |
| | 32 | **27.8** | **(8,8)** | **3.8** | 27.8 | (8,8) | 5.3 | **27.8** | **(8,8)** | **3.8** |
| **GPT2-137M** (Float: 29.9) | 16 | - | - | - | 3345.8 | (3,3) | 93.2 | **552.4** | **(3,6)** | **55.4** |
| | 17 | - | - | - | 2705.3 | (3,6) | 75.1 | **310.1** | **(3,7)** | **52.8** |
| | 18 | 3760.3 | (3,3) | 82.3 | 1100.5 | (4,5) | 52.9 | **134.3** | **(4,7)** | **34.9** |
| | 19 | 2782.2 | (3,4) | 43.9 | 402.9 | (4,6) | 47.3 | **67.5** | **(4,7)** | **25.6** |
| | 20 | 742.4 | (3,5) | 55.3 | 213.2 | (4,7) | 44.3 | **40.4** | **(4,8)** | **24.5** |
| | 21 | 356.2 | (3,6) | 48.8 | 85.2 | (5,7) | 24.9 | **33.2** | **(5,8)** | **13.2** |
| | 22 | 189.9 | (4,6) | 26.4 | 46.3 | (5,8) | 23.8 | **32.1** | **(6,8)** | **7.3** |
| | 23 | 65.8 | (4,7) | 24.7 | 34.2 | (6,8) | 13.0 | **31.8** | **(6,8)** | **6.3** |
| | 24 | 39.8 | (4,8) | 23.8 | 32.1 | (7,8) | 7.1 | **31.5** | **(7,8)** | **3.2** |
| | 32 | **31.5** | **(8,8)** | **1.6** | 31.6 | (8,8) | 3.2 | **31.5** | **(8,8)** | **1.6** |
| **Pythia-160M** (Float: 26.7) | 16 | - | - | - | 4501.1 | (3,4) | 76.8 | **386.0** | **(3,6)** | **53.2** |
| | 17 | - | - | - | 3095.1 | (3,5) | 72.5 | **198.6** | **(3,6)** | **46.3** |
| | 18 | 9887.1 | (3,3) | 49.4 | 1070.2 | (4,5) | 46.7 | **74.5** | **(4,6)** | **25.1** |
| | 19 | 1946.8 | (3,4) | 49.8 | 391.7 | (4,6) | 42.9 | **46.2** | **(4,7)** | **24.4** |
| | 20 | 456.2 | (3,5) | 47.8 | 117.5 | (5,6) | 23.6 | **34.6** | **(5,7)** | **13.3** |
| | 21 | 198.3 | (3,6) | 45.1 | 78.5 | (5,7) | 23.4 | **32.4** | **(5,8)** | **13.3** |
| | 22 | 69.6 | (4,6) | 23.5 | 48.6 | (5,7) | 21.2 | **30.1** | **(6,8)** | **7.8** |
| | 23 | 44.4 | (4,7) | 22.6 | 37.2 | (6,8) | 13.0 | **28.2** | **(6,8)** | **5.5** |
| | 24 | 33.2 | (5,7) | 11.3 | 31.8 | (7,8) | 7.4 | **27.6** | **(7,8)** | **2.8** |
| | 32 | **27.4** | **(8,8)** | **1.4** | 27.5 | (8,8) | 2.7 | **27.4** | **(8,8)** | **1.4** |

Table 8: **OPTQ:** We provide the test perplexity (PPL) and quantization configuration of the Pareto-optimal models that form the frontiers visualized in Figure 1. Note that $M$ and $N$ respectively denote the weight and activation bit widths.

| Model | P | OPTQ | | | OPTQ+EP-init | | | OPTQ+AXE | | |
|---|---|---|---|---|---|---|---|---|---|---|
| | | **PPL** | $(M,N)$ | **Sparsity** | **PPL** | $(M,N)$ | **Sparsity** | **PPL** | $(M,N)$ | **Sparsity** |
| **OPT-125M** (Float: 27.7) | 16 | - | - | - | 3333.8 | (4,5) | 62.2 | **225.0** | **(3,6)** | **52.8** |
| | 17 | - | - | - | 1722.6 | (4,5) | 53.6 | **80.2** | **(3,6)** | **45.7** |
| | 18 | 9942.5 | (3,3) | 54.5 | 409.8 | (5,5) | 36.1 | **41.3** | **(4,6)** | **26.6** |
| | 19 | 8278.3 | (3,4) | 47.5 | 136.0 | (5,6) | 35,7 | **35.0** | **(5,6)** | **15.1** |
| | 20 | 281.1 | (3,5) | 45.5 | 46.9 | (5,6) | 26.8 | **31.3** | **(5,6)** | **14.2** |
| | 21 | 60.4 | (3,6) | 44.7 | 40.1 | (5,7) | 26.8 | **29.0** | **(5,7)** | **14.2** |
| | 22 | 35.7 | (4,6) | 25.8 | 30.3 | (6,7) | 15.6 | **28.5** | **(5,8)** | **14.2** |
| | 23 | 31.5 | (5,6) | 14.6 | 29.7 | (6,8) | 15.6 | **28.0** | **(6,8)** | **8.6** |
| | 24 | 29.2 | (5,7) | 14.6 | 28.1 | (7,8) | 9.5 | **27.8** | **(7,8)** | **5.4** |
| | 32 | **27.8** | **(8,8)** | **2.2** | 27.8 | (8,8) | 5.6 | 27.8 | (8,8) | 2.2 |
| **GPT2-137M** (Float: 29.9) | 16 | - | - | - | 2765.6 | (4,4) | 52.6 | **1513.6** | **(4,5)** | **34.0** |
| | 17 | - | - | - | 2465.0 | (4,4) | 49.0 | **496.4** | **(3,6)** | **43.4** |
| | 18 | 4140.7 | (3,3) | 59.3 | 2465.0 | (4,4) | 49.0 | **117.9** | **(4,6)** | **24.2** |
| | 19 | 2782.2 | (3,4) | 43.9 | 1108.4 | (5,6) | 34.5 | **59.9** | **(4,7)** | **24.2** |
| | 20 | 2149.8 | (4,4) | 26.0 | 361.7 | (4,7) | 43.6 | **45.5** | **(5,7)** | **13.1** |
| | 21 | 1153.8 | (4,5) | 24.7 | 73.1 | (5,7) | 24.7 | **37.3** | **(5,8)** | **13.2** |
| | 22 | 176.9 | (4,6) | 24.0 | 42.7 | (5,8) | 24.5 | **33.1** | **(6,8)** | **12.2** |
| | 23 | 50.1 | (4,7) | 23.2 | 33.5 | (6,8) | 13.4 | **32.1** | **(6,8)** | **6.2** |
| | 24 | 37.4 | (5,7) | 12.2 | 32.0 | (7,8) | 7.3 | **31.8** | **(7,8)** | **3.1** |
| | 32 | 31.8 | (8,8) | 1.6 | **31.7** | **(8,8)** | **3.3** | 31.7 | (8,8) | 1.6 |
| **Pythia-160M** (Float: 26.7) | 16 | - | - | - | 6739.6 | (4,6) | 79.7 | **1521.2** | **(3,5)** | **41.7** |
| | 17 | - | - | - | 5345.7 | (4,5) | 49.9 | **311.7** | **(4,5)** | **22.9** |
| | 18 | 27098.1 | (3,3) | 40.5 | 1372.4 | (4,5) | 41.1 | **126.1** | **(4,6)** | **23.1** |
| | 19 | 5644.0 | (3,4) | 40.3 | 641.2 | (4,6) | 41.0 | **61.4** | **(4,6)** | **21.3** |
| | 20 | 948.4 | (3,5) | 40.1 | 132.9 | (5,6) | 23.4 | **43.5** | **(5,6)** | **10.9** |
| | 21 | 151.3 | (4,5) | 21.4 | 108.5 | (5,7) | 23.5 | **32.8** | **(5,7)** | **10.9** |
| | 22 | 61.4 | (4,6) | 21.3 | 74.1 | (5,7) | 22.0 | **30.0** | **(5,8)** | **10.9** |
| | 23 | 43.3 | (5,6) | 10.9 | 40.4 | (6,8) | 13.0 | **28.0** | **(6,8)** | **5.5** |
| | 24 | 32.8 | (5,7) | 10.9 | 32.1 | (7,8) | 7.5 | **27.4** | **(7,8)** | **2.7** |
| | 32 | **27.2** | **(8,8)** | **1.4** | 27.6 | (8,8) | 2.9 | 27.2 | (8,8) | 1.4 |

