# OpenReview forum: "Accumulator-Aware Post-Training Quantization for Large Language Models"
_ICLR.cc/2025/Conference — Submitted to ICLR 2025_

### Official Review · Reviewer_2viG · 2024-10-30

**Soundness:** 3
**Presentation:** 2
**Contribution:** 2
**Rating:** 5
**Confidence:** 3

**Summary:**

This paper introduces AXE, a framework for overflow-avoidant, accumulator-aware quantization in the PTQ setting, aimed at efficient low-precision deployment of large models. AXE extends existing PTQ methods like GPFQ and OPTQ by managing accumulator bit width to improve resource efficiency without requiring retraining, and supports multi-stage accumulation for scaling to large models. The authors report gains in task performances over PTQ baselines with reduced accumulator bit width.

**Strengths:**

1. The AXE method is theoretically grounded.

2. AXE fills a gap in PTQ by addressing overflow risks in low-precision settings, potentially benefiting deployment on resource-constrained hardware.

3 The paper introduces a novel overflow handling approach in PTQ, potentially expanding PTQ’s applicability to larger models.

**Weaknesses:**

1. I didn't find any evidence of actual efficiency gain in the experiments apart from the bit width reduction and perplexity improvements.

2. Currently the paper evaluates on perplexity and zero-shot reasoning tasks, this IMO is not enough, more NLP tasks need to be tested to consolidate the efficiency of the AXE method.

3. Novelty: While AXE extends existing PTQ methods with accumulator-aware quantization, much of its methodology relies on previously established concepts. The theoretical contributions build on recent QAT-based methods, and the extension to PTQ, while practical, does not introduce a fundamentally new approach to quantization beyond overflow handling.

4. Writing could be improved, in particular the topic introduction and storytelling. I had a hard time following the paper.

**Questions:**

What does Table 4 try to deliever here? There is no description in the main context of the paper.

---

> ### Author Response · Authors · 2024-11-19
>
> Thank you for your question. We respectfully direct you to Section 4.1 (lines 485-501), where we reference Appendix C.2 as an ablation study targeting a monolithic accumulator. In Appendix C.2 (lines 929-945), we provide further details and interpretations of the results from Table 4. From these results, we conclude multi-stage accumulation improves scaling to increasingly large language models.
> We acknowledge that there is no explicit mention of Table 4 in the body of the text, although we encourage you to read the caption of the table and its preceding paragraph. If you find it helpful, we will make explicit mention in the body of Appendix C.2 to improve readability upon acceptance.
>
> We humbly request more actionable feedback. We would appreciate the opportunity to directly address your concerns in accepting this paper. Evaluating the quality of language models is a rich and rapidly developing field in and of itself. Thus, language model quantization research has an established precedent of investigating how bit width reduction impacts both perplexity and zero-shot (or few-shot) reasoning tasks (please see OPTQ and SmoothQuant as examples). In addition, accumulator-aware post-training quantization is fundamentally a new quantization paradigm; weight-only quantization is incapable of accumulator-awareness by construction (please see Section 3.3, lines 356-366) and weight-activation quantization is insufficient (please see Figure 1).
> In this work, we formulate the accumulator-aware PTQ paradigm objective and propose an approximate solution with theoretical justification. In such a paradigm, the accumulator bit width is a new dimension to be considered in the quantization design space alongside weight and activation bit widths. AXE marks the first solution that: (1) enables the co-optimization of a quantized model and its target datapath; and (2) scales to billion-parameter LLMs. We respectfully direct you to Section 2.2 (lines 141-150), Section 3.3 (lines 378-390), and our response to Reviewer SPYS, where we highlight the established benefits of this co-optimization opportunity.

---

### Official Review · Reviewer_wtrT · 2024-10-31

**Soundness:** 3
**Presentation:** 2
**Contribution:** 2
**Rating:** 5
**Confidence:** 4

**Summary:**

This paper tackles a specific challenge related to low-level hardware architecture, specifically targeting the accumulation process in a Post-Training Quantization (PTQ) setting. The accumulator design in current hardware architectures is prone to significant numerical deviations when numbers are heavily quantized. The paper examined two distinct cases: a singular accumulator and an accumulator within an adder tree's output. The analysis largely focuses on integer arithmetic. The paper's core objective is to enhance a subset of quantization algorithms, such as GPFQ, by proposing an L1-norm penalty for high-magnitude post-quantization weights, as these can introduce errors in subsequent accumulation stages.

**Strengths:**

The paper is well-written and easy to understand. The optimization it proposes concentrates on low-level hardware details that significantly differs from existing approaches in quantization research. Notably, the issue of accumulation round-off errors, which the paper addresses, is frequently overlooked by the Efficient AI community.

**Weaknesses:**

The paper is too focused on quatnization that is associated with the low-level hardware architecture, making me feel ICLR may not be a very suitable venue for work like this.

The paper's presentation raises concerns regarding its background setup and evaluation.

First, it fails to acknowledge a range of prior studies in this field, including LLM.int8, ZeroQuant, AWQ, and others. Moreover, the paper lacks comparative analysis with weight-activation quantization methods, making it difficult to gauge the effectiveness of the proposed technique relative to existing quantization research. Most current works aim to quantize weight values to alleviate pressure on HBM bandwidth. Typically, they dequantize model parameters and perform multiply-accumulate operations in higher precision (e.g., fp16). Without a clear comparative study on end-to-end GPU performance, it is challenging to understand the benefits of the suggested method, especially in memory-bound LLM inference, and how much advantage is gained from enabling accurate low-precision arithmetic operations.

Second, the proposed method is mainly evaluated on a single modern LLM family (Pythia). The model under the choice here (Pythia) is not a popular one, and this again, makes comparing to other methods very challenging. An obvious good candidate for an evaluation like this would be the LLaMA family models. The paper also extends the evaluation to the OPT and GPT2 models. However, both models are fairly small in size (< 1B) and also are fairly old.

**Questions:**

Based on the two points I raised in Weakness, can you please

1. Add actual performance metrics (eg.  throughput) in Table 1 and a direct comparison to more related method that are actually integrated in LLM serving engines (vLLM)
2. Extend the accuracy comparison to more LLM model famil

---

> ### Author Response · Authors · 2024-11-19
>
> Thank you for your feedback and questions. To conserve space, we address Q1(a) in our response to Reviewer SPYS, where we highlight Ni et al. (2021) (published in ICLR) as an exemplar analysis of the benefits of low-precision accumulation for ASICs. We appreciate your understanding and hope the following will address your remaining concerns in accepting this paper.
>
> Re: Q1(b), it is an artifact of recent naming conventions and benchmarking practices that techniques such as AWQ and SmoothQuant are perceived as mutually exclusive to techniques such as OPTQ and GPFQ; in reality these techniques are composable, as we additionally discuss in our response to Reviewer xjjZ. We respectfully direct the Reviewer to Appendix C.1 (lines 891-901), where we describe how we compose our baselines with SmoothQuant and bias correction, which are complementary techniques that can respectively be applied before and after the chosen greedy sequential quantization algorithm (e.g., OPTQ). We highlight that, aside from the target datatypes of their study, the primary difference between the SmoothQuant and AWQ algorithms is that SmoothQuant tunes the scaling factors $s$ with a fixed $\alpha$ while AWQ tunes $\alpha$ with a fixed heuristic for $s$, otherwise they approximately solve the same objective function. With the goal of our research being to investigate the feasibility of reducing the accumulator bit width in the PTQ setting, we use established PTQ methods that solve orthogonal problems with the intention to create high quality reference baselines. Note that we acknowledge in Section 4.1 (lines 445-446) that our experiments do not directly evaluate the original GPFQ and OPTQ proposals.
>
> Re: Q2, we admit that more models could always be considered, although a direct comparison against more quantization methods would be orthogonal as no other PTQ techniques have accumulator-awareness, which is the focus of our study. To the best of our knowledge, only datatype manipulation and EP-init serve as alternatives to AXE for accumulator-aware quantization in the PTQ setting (please see Figure 1 for a comparison). Our intention with focusing on Pythia was to investigate scaling, for which the Pythia model family was specifically designed, as discussed in their well-cited study. However, to demonstrate both generalization to another LLM family as well as the composition of quantization techniques, we provide additional results below with Llama3 on WikiText2. We use the same W4A8 quantization scheme discussed in Appendix C aside from using a unique scaling factor per-token rather than per-tensor, which improves performance without impacting our analysis. We also use the same quantization process described in Appendix C aside for removing bias correction, which seems to have minimal impact on these models from this preliminary analysis. In the spirit of AWQ, we perform a light grid search over SmoothQuant's $\alpha$ parameter and find 0.4 to generally perform the best on average for these models. We highlight that constraining Llama3 models to 16-bit accumulation with AXE with OPTQ and SmoothQuant generally outperforms the unconstrained OPTQ baseline. We would be happy to include these results in the appendix if you find it helpful. Either way, we will release open-source reference implementations for reproducibility.
>
> |                              | Llama3.1-1B | Llama3.2-3B | Llama3.2-8B |
> |------------------------------|-------------|-------------|-------------|
> | **FP16**                     | 11.78       | 9.06        | 6.47        |
> | **OPTQ**                     | 37.00       | 12.82       | 7.39        |
> | **OPTQ + SmoothQuant**       | 14.47       | 10.20       | 7.48        |
> | **OPTQ + SmoothQuant + AXE(128x32b)** | 14.43       | 10.24       | 7.47        |
> | **OPTQ + SmoothQuant + AXE(128x16b)** | 14.90       | 10.45       | 7.61        |

---

> > ### Comment · Reviewer_wtrT · 2024-11-23
> >
> > My concern is that most weight-only quantization methods do not take into account accumulator-awareness for a valid reason—typically, they rely on loading quantized weight parameters and subsequently dequantize them for high-precision accumulation in floating-point. This is because the inference bottleneck is often the bandwidth of High Bandwidth Memory (HBM). Given this context, I question the necessity of accumulating in the format you propose, especially when I'm willing to spend additional computational resources on dequantization and perform later compute in high-precison, considering that model inference is memory-bound rather than compute-bound. Therefore, I believe you cannot simply state that your method is entirely orthogonal without further justification. I hope I made my my point clearer here.
> >
> > I also observed that the added results do not seem to exhibit a significant improvement over the baseline.

---

> > > ### Author Response · Authors · 2024-11-26
> > >
> > > Thank you for your engagement. We largely agree with your assessment that low-precision integer accumulation should provide the most benefits on compute-bound workloads. We hope our response to your comment on the thread with Reviewer SPYS provides further clarity on the topic of inference benefits. Below, we highlight the improvements in model quality via AXE.
> > >
> > > The Llama3 results we shared were intended to demonstrate that our scaling analysis can be extended to more recent model families. As such, they were organized to show that various techniques are composable, and our constraints were able to maintain performance relative to an unconstrained baseline. We emphasize that only datatype manipulation and EP-init serve as alternatives to AXE for accumulator-aware quantization in the PTQ setting. As shown in Figure 1, AXE is a Pareto-dominant algorithm. Moreover, as discussed in Section 3.2 (lines 337-338), AXE has the desired feature of being functionally equivalent to the underlying algorithm (e.g., OPTQ or GPFQ) when the accumulator is large enough (e.g., 32 bits). Thus, one should expect these benefits to manifest when targeting low-precision accumulators (e.g., 16 bits) and not high-precision accumulators (e.g., 32 bits).
> > >
> > > That said, your questioning raises a valid point on establishing a baseline for these Llama3 results. Our experiments suggest that multi-stage accumulation is critical to enable low-precision accumulation on increasingly large language models (please see Appendix C.2). Note that EP-init does not support multi-stage accumulation. Yet, as an alternative to AXE, one can still guarantee overflow avoidance in tiles via datatype manipulation; W4A4 guarantees overflow avoidance for 16-bit accumulation in tiles of 128 elements (i.e., 128x16b). With this, please find a more complete analysis of Llama3 on WikiText2 below, where we use "inf" to denote a perplexity over 1000.
> > >
> > > | Configuration | Method                | 3.2-1B | 3.2-3B | 3.1-8B |
> > > |---------------|-----------------------|--------|--------|--------|
> > > | 128x32b       | OPTQ + SQ(0.4)        | 14.5   | 10.2   | 7.5    |
> > > | 128x32b              | OPTQ + SQ(0.4) + AXE  | 14.4   | 10.2   | 7.5    |
> > > | 128x16b       | OPTQ + SQ(0.4)        | inf    | inf    | inf    |
> > > |  128x16b             | OPTQ + SQ(0.4) + AXE  | 14.9   | 10.4   | 7.6    |
> > >
> > > Note that not only does AXE enable low-precision accumulation for Llama3 with minimal degradation from the unconstrained baseline (i.e., 128x32b), but the gap between the constrained an unconstrained baselines decreases as the model size increases. This result aligns with our theoretical analysis on the Pythia models (Section 4.2).

---

### Official Review · Reviewer_SPYS · 2024-11-03

**Soundness:** 2
**Presentation:** 3
**Contribution:** 2
**Rating:** 6
**Confidence:** 2

**Summary:**

The paper is the first study on post-training quantization while keeping the accumulator in the picture. A low-bit accumulator risks to overflow, but speeds up the arithmetic computation. They use L1 constraint on the inner product counterpart to guarantee numerical stability in equation (2) and build a quantization method that controls the accumulator based on this result.
The L1 constraint is the same formulation as first introduced in compressed sensing by Donoho (1994) and later called the lasso of Tibshirani (1996) in the context of linear regression.

**Strengths:**

Thi sis the first formal study of quantization on the accumulator size. The paper is well-written and easy to follow with theoretical justifications. The innovative idea appears in equation (17) as a layer-wise operation. The authors adapt this result for two well-known post-training quantization methods GPFQ, and OPTQ.

**Weaknesses:**

Although this is the first study on accumulators, I doubt its usefulness.
Often, the accumulator size is hardware-dependent, and sometimes even unknown. There are ways to guess the accumulator size by running various experiments, but they are not revealed by the manufacturer. In the context of quantization only weights or weight-activation, the benefit is clear; I wonder how we can benefit from quantizing accumulators unless we design a new processor or a co-processor. This limits the impact of this study unless the authors provide a guideline on how to use the accumulator bit size in practice on certain processor.

**Questions:**

- CPU cores, cude cores, tensor cores support different number formats, each with different (unknown) accumulator size. How quantizing accumulator lead to advantages for pre-training, fine-tuning, or inference of LLM models?
- Using OPTQ and GFPQ as a baseline is interesting in terms of accuracy, but they lead to smaller models after their use. What AXE brings to the table practically?

---

> ### Author Response · Authors · 2024-11-19
>
> Thank you for your questions and feedback. We acknowledge that a deeper discussion on inference benefits would be useful; Reviewers xjjZ and wtrT have similarly raised this important question. We hope the following will address your concerns in accepting this paper.
>
> Just as the benefits of reducing weight and activation bit widths are non-uniform across platforms (e.g., 3-bit compute is hard to accelerate on CPUs), the benefits of reduced accumulator bit widths often depend on some combination of compiler-level software support, instruction sets of the target platform, and availability of suitable datapaths in hardware. Many prior works have motivated accumulator bit width reduction by characterizing its benefits across various dimensions on different platforms. For FPGAs, Colbert et al. (2023) show that reducing the accumulator bit width improves area efficiency by reducing on-device memory requirements and datapath width. For CPUs, Xie et al. (2021) show that reducing the accumulator bit width can increase hardware utilization to improve both throughput and bandwidth efficiency. We briefly discuss these prior works in Section 2.2 (lines 141-150) and Section 3.3 (lines 378-390). We respectfully highlight Ni et al. (2021), published in ICLR, as an exemplar analysis of the power, area, and throughput benefits on ASICs. Given this wide range of established benefits, it was our intention to use our limited page count to directly extend the scope of accumulator-aware weight quantization to the PTQ setting from the perspective of applied theory, effectively decoupling our own investigations from the capabilities of any one target platform. Our work formulates the accumulator-aware PTQ objective and proposes AXE as an approximate solution, generally enabling future researchers or practitioners to capitalize on the proven benefits of low-precision accumulation without requiring end-to-end retraining. However, we agree a deeper investigation focused on the benefits of low-precision accumulation in LLM inference would be an impactful study if it could augment the well-established benefits with new insights.

---

> > ### Comment · Reviewer_SPYS · 2024-11-22
> > **More detailed comparison**
> >
> > I request that the authors provide a more detailed comparison of the practical benefits of AXE over OPTQ and GPFQ.
> > Please provide detailed information about inference speed (latency), energy efficiency, or memory usage in addition to model size and accuracy.

---

> > > ### Author Response · Authors · 2024-11-23
> > >
> > > We appreciate your request for a more detailed comparison of the practical benefits of AXE over OPTQ and GPFQ. Below, we provide a high-level theoretical analysis, supplemented by specific examples, to clarify the advantages of incorporating low-precision accumulation into inference workflows.
> > >
> > > First, consider the computational complexity of an operation such as $ \textstyle \sum_{i=1}^N w_i x_i $. If the complexity of multiplying two $b$-bit numbers is $m(b)$ and that of adding two $c$-bit numbers is $a(c)$, the overall complexity of this sum can be approximated as $N m(b) + (N-1) a(c)$. Replacing an accumulator of $c=32$ bits with $c=16$ bits would be expected to reduce the term’s contribution to the overall complexity by roughly a factor of two as the complexity of addition typically scales linearly with $c$ (i.e., $O(c)$) [1]. Furthermore, as $b$ is reduced, one can expect this reduction in complexity to yield more significant savings via Amdahl’s Law since $m(b)$ scales quadratically with $b$ [1]. Thus, one can expect low-precision accumulation to become even more important as weight and activation bit widths are reduced.
> > >
> > > A more detailed analysis would depend heavily on the specifics of the data pipeline and hardware architecture. However, these “back-of-the-envelope” estimates align with published experimental reports on low-precision accumulation. For example, assume either OPTQ or GPFQ are used to quantize ResNet18 to W8A8 and the resulting model is then deployed to a platform supporting both 32-bit and 16-bit accumulation. When using AXE to target 16-bit accumulation instead of 32-bit accumulation, one can expect 1.5x and 1.6x latency improvements on an MTK8167s or Allwinner V328 ARM processor, respectively [2]. If 8-bit accumulation is also supported, one can also expect up to 2.5x latency improvements when using AXE to instead target 8-bit accumulation on an Intel i7 CPU [3].
> > >
> > > With respect, a deep analysis of the unique benefits of low-precision accumulation in an end-to-end LLM inference pipeline is outside the scope of our research. It is likely an entirely new study in its own right and may require novel low-level kernel optimizations to capture these theoretical benefits. However, via logical deduction, we expect low-precision accumulation to provide greater benefits in the prefill stage than the decoding stage in the LLM inference pipeline. Low-precision accumulation is known to provide more significant speedups on compute-bound workloads [4]. As the prefill stage is commonly compute-bound, we expected the aforementioned latency improvements to directly translate to throughput improvements. However, the decode stage is often memory-bound and may limit the peak attainable uplift provided from low-precision accumulation.
> > >
> > >
> > > [1] Baskin, Chaim, et al. "Uniq: Uniform noise injection for non-uniform quantization of neural networks." ACM Transactions on Computer Systems (TOCS) 37.1-4 (2021): 1-15.
> > >
> > > [2] Xie, Hongwei, et al. "Overflow aware quantization: Accelerating neural network inference by low-bit multiply-accumulate operations." Proceedings of the Twenty-Ninth International Conference on International Joint Conferences on Artificial Intelligence. 2021.
> > >
> > > [3] Ni, Renkun, et al. "Wrapnet: Neural net inference with ultra-low-precision arithmetic." International Conference on Learning Representations ICLR 2021. OpenReview, 2021.
> > >
> > > [4] [Open-sourcing FBGEMM for server-side inference - Engineering at Meta](https://engineering.fb.com/2018/11/07/ml-applications/fbgemm/)

---

> > > > ### Comment · Reviewer_wtrT · 2024-11-23
> > > >
> > > > Apologies cutting into this thread, but I believe my concern is somewhat connected to this discussion. Would it be correct to say that the advantages of your proposed method primarily manifest when performing low-precision matrix multiplication in integer form?
> > > >
> > > > However, targeting this optimization may not be useful in memory-bound scenarios, which is normally the case for LLM inference. While I recognize there may be applications for LLMs in devices like AI PCs, where the proposed method becomes very relavent, I find it difficult to see the practicality of this method for cloud-based platforms where normally we have a big FLOPs/second number. Do you really have a runtime/throughput/energy efficiency benefits on these platforms compared to existing methods?
> > > >
> > > > I believe it's crucial to specify the intended use-case scenarios and clarify the limitations for your readers. Respectfully, I must express my skepticism regarding the framing of your approach as an orthogonal optimization to existing methods. While accumulation quantization optimization is indeed orthogonal in nature as an optimizaiton problem. However, it might prove to be irrelevant if the performance is not constrained by compute, allowing me to afford the luxury of performing high-precision accumulation without these treatments.

---

> > > > > ### Author Response · Authors · 2024-11-26
> > > > > **On the inference benefits of low-precision accumulations (1/2)**
> > > > >
> > > > > Thank you for your engagement and the precision of your phrasing. We are converging on a common reference frame with which to communicate these ideas. We agree that while accumulator-aware quantization is orthogonal in nature as a theoretical optimization problem, it is not orthogonal as an end-to-end inference optimization problem. With this shared understanding, we more carefully discuss our hypotheses for end-to-end inference below. However, we emphasize that the focus of our paper is applied theory. We expect a more detailed theoretical and experimental analysis of end-to-end inference optimization to be its own study that may yield new insights. We hope that you and the other reviewers allow space for applied quantization theory in this conference. We firmly believe our theoretical contributions, which are the first to our knowledge to enable accumulator-aware quantization on LLMs, are a critical step towards end-to-end optimization for LLM inference. Moreover, cloud-based LLM serving is not the only use case. You rightfully point to AI PCs; however, there exists more opportunities such as increased area efficiency in streamlined inference accelerators on FPGAs [1] and increased energy efficiency for resource-constrained edge platforms with custom ASICs [2,3].
> > > > >
> > > > > That said, your point is well-taken. We largely agree with your assessment that low-precision integer accumulation should provide the most benefits on compute-bound workloads, but there may be more nuance. We expect benefits to manifest elsewhere as we will show with a few examples below.
> > > > >
> > > > > First, as you have rightfully pointed out, memory bottlenecks are a critical challenge in hyper-scale LLMs. It has been the focal point of a large body of research in the past few years, from algorithm optimizations (e.g., FlashAttention) to network architecture (e.g., SSMs). Ultimately, as this research continues to reduce memory bottlenecks, one can expect more language models to ultimately become compute bound. Thus, in the future, more models should benefit from low-precision accumulation and, therefore, our accumulator-aware PTQ framework.
> > > > >
> > > > > Second, one can justify memory bottlenecks rendering latency improvements irrelevant when the communication overhead ultimately masks the computation work. However, unlike time costs, energy costs cannot be masked via parallelization. For an excellent theoretical analysis of this, we reference the Roofline Model of Energy proposed by Choi et al. (2013) [4], where the energy cost of compute and memory are added to the constant energy required to perform an operation. Note that this aligns with our earlier logical deduction (restated below for convenience). We also reference the seminal paper by Horowitz (2014) [5], which reports 32-bit integer additions to be ~3x more expensive than 8-bit additions and 32-bit multiplications to be ~15x more expensive than 8-bit multiplications, which aligns with our linear and exponential scaling projections. Thus, we can expect these energy savings to become more important as weight and activation bit widths are reduced and overall memory traffic is reduced.
> > > > >
> > > > > > First, consider the computational complexity of an operation such as $ \textstyle \sum_{i=1}^N w_i x_i $. If the complexity of multiplying two $b$-bit numbers is $m(b)$ and that of adding two $c$-bit numbers is $a(c)$, the overall complexity of this sum can be approximated as $N m(b) + (N-1) a(c)$. Replacing an accumulator of $c=32$ bits with $c=16$ bits would be expected to reduce the term’s contribution to the overall complexity by roughly a factor of two as the complexity of addition typically scales linearly with $c$ (i.e., $O(c)$) [6]. Furthermore, as $b$ is reduced, one can expect this reduction in complexity to yield more significant savings via Amdahl’s Law since $m(b)$ scales quadratically with $b$ [6]. Thus, one can expect low-precision accumulation to become even more important as weight and activation bit widths are reduced.
> > > > >
> > > > > [1] Colbert et al. "A2Q: Accumulator-aware quantization with guaranteed overflow avoidance." Proc. of the IEEE/CVF International Conference on Computer Vision. 2023.
> > > > >
> > > > > [2] Ni et al. "Wrapnet: Neural net inference with ultra-low-precision arithmetic." International Conference on Learning Representations ICLR 2021.
> > > > >
> > > > > [3] Azamat et al. "Squeezing accumulators in binary neural networks for extremely resource-constrained applications." Proc. of the 41st IEEE/ACM International Conference on Computer-Aided Design. 2022.
> > > > >
> > > > > [4] Choi et al. "A roofline model of energy." 2013 IEEE 27th International Symposium on Parallel and Distributed Processing. IEEE, 2013.
> > > > >
> > > > > [5] Horowitz, Mark. "1.1 computing's energy problem (and what we can do about it)." 2014 IEEE international solid-state circuits conference digest of technical papers (ISSCC). IEEE, 2014.
> > > > >
> > > > > [6] Baskin et al. "Uniq: Uniform noise injection for non-uniform quantization of neural networks." ACM Transactions on Computer Systems (TOCS) 37.1-4 (2021): 1-15.

---

> > > > > ### Author Response · Authors · 2024-11-26
> > > > > **On the inference benefits of low-precision accumulations (2/2)**
> > > > >
> > > > > Finally, we do expect cloud-based platforms to realize benefits, possibly even more so in some cases. Cloud providers can batch requests to increase data reuse which may mitigate the overhead of data transfer and allow workloads to more consistently benefit from low-precision accumulation. To provide some supporting evidence on the benefits of low-precision accumulation on data center GPUs, we slightly modify the matrix multiplication (matmul) kernel from the Triton tutorial [1] to compare `int32` and `int16` accumulation on an illustrative microbenchmark. In this admittedly narrow demonstration, we use `BLOCK_SIZE_K=128` as this was what we assume in the scaling analysis in our paper (i.e., $T=128 \times 16$b) and fix the compute tiling such that `BLOCK_SIZE_M = BLOCK_SIZE_N = 256`. We recognize that a practitioner may allow the Triton autotuner to optimize over the block sizes; however, we pin these parameters to limit their influence on the following analysis. We then look at the dimensions of Pythia-1B; the QKV matmuls are ($BS\times2048$) x ($2048\times2048$), the FFN-Up matmuls are ($BS\times2048$) x ($2048\times8192$), and the FFN-Down matmuls are ($BS\times8192$) x ($8192\times2048$), where $B$ is the batch size and $S$ is the sequence length. We experiment with $B \in { 1, 32 \} $ and $S \in \{32, 2048 \}$. We use `int8` inputs and compare `int32` against `int16` accumulation on an AMD MI210, a high-performance data center GPU. We report the median latency observed over 300 samples in the table below. We also provide $<M, N, K>$ as they are labeled in the Triton tutorial. All latency results are in milliseconds, and we provide the relative uplift (i.e., rel).
> > > > >
> > > > > | B, S       | Layer    | M, K, N              | int32  | int16  | rel  |
> > > > > |------------|----------|----------------------|--------|--------|------|
> > > > > | B=1, S=32  | QKV      | M=BS, K=2048, N=2048 | 1.04   | 1.05   | 0.99 |
> > > > > |            | FFN-Up   | M=BS, K=2048, N=8192 | 1.32   | 1.29   | 1.02 |
> > > > > |            | FFN-Down | M=BS, K=8192, N=2048 | 1.90   | 1.87   | 1.02 |
> > > > > | B=1, S=2048| QKV      | M=BS, K=2048, N=2048 | 7.68   | 6.91   | 1.11 |
> > > > > |            | FFN-Up   | M=BS, K=2048, N=8192 | 30.23  | 27.11  | 1.12 |
> > > > > |            | FFN-Down | M=BS, K=8192, N=2048 | 28.72  | 25.50  | 1.13 |
> > > > > | B=32, S=32 | QKV      | M=BS, K=2048, N=2048 | 4.13   | 3.77   | 1.10 |
> > > > > |            | FFN-Up   | M=BS, K=2048, N=8192 | 15.17  | 13.66  | 1.11 |
> > > > > |            | FFN-Down | M=BS, K=8192, N=2048 | 14.39  | 12.80  | 1.12 |
> > > > > | B=32, S=2048| QKV     | M=BS, K=2048, N=2048 | 242.66 | 215.94 | 1.12 |
> > > > > |            | FFN-Up   | M=BS, K=2048, N=8192 | 985.49 | 868.93 | 1.13 |
> > > > > |            | FFN-Down | M=BS, K=8192, N=2048 | 965.00 | 841.28 | 1.15 |
> > > > >
> > > > >
> > > > > Please note that this is not a formal performance comparison of `int32` vs. `int16` accumulation. We make minor changes in the Python interface for the Triton compiler, which then generates MLIR, which is then compiled to AMD ISA. According to AMD ISA reference guide [2], 16-bit accumulation is not natively supported for the MI200 architecture. From a cursory inspection of the kernel disassembly, we conjecture that the benefits are from improvements on cache behavior and not compute acceleration. Therefore, these results may actually be an underestimate of the potential benefits of low-precision accumulation in cloud-based LLM applications. Intuitively, the table suggests that increased data reuse yields further improvements for lower precision accumulators even without a 16-bit accumulator datapath. Therefore, it would be reasonable to assume hardware with native 16-bit accumulator support would yield further uplift.
> > > > >
> > > > > As with the Llama3 results in the other thread, we are happy to open-source reference code after the review process.
> > > > >
> > > > > [1] [Matrix Multiplication — Triton documentation](https://triton-lang.org/main/getting-started/tutorials/03-matrix-multiplication.html)
> > > > >
> > > > > [2] [AMD Instinct MI200 Instruction Set Architecture: Reference Guide](https://www.amd.com/content/dam/amd/en/documents/instinct-tech-docs/instruction-set-architectures/instinct-mi200-cdna2-instruction-set-architecture.pdf)

---

> > > > ### Comment · Reviewer_SPYS · 2024-11-26
> > > > **Feedback to complexity analysis**
> > > >
> > > > I would like to thank the authors for taking the time to provide a complexity analysis for accumulator size.
> > > >
> > > > Their complexity analysis only shows the potential of lowering the accumulator size in terms of speed, if properly implemented in hardware. I  hope the authors highlight the benefit of their method in practice through numbers evaluated in an experimental setting, or in a hardware emulator.

---

> > > > > ### Author Response · Authors · 2024-11-26
> > > > >
> > > > > Thank you, Reviewer SPYS, for your acknowledgement of our complexity analysis. We humbly direct your attention to our 2-part response to Reviewer wtrT titled **On the inference benefits of low-precision accumulations**, where we provide experimental numbers for a microbenchmark as well as a deeper analysis on energy benefits.
> > > > >
> > > > > If you and Reviewer wtrT find this analysis sufficient and insightful, we can quickly update the submission with this preliminary analysis. Thank you for your engagement.

---

> > > > > > ### Comment · Reviewer_SPYS · 2024-11-26
> > > > > > **Reply to Authors comments on numerical evaluation**
> > > > > >
> > > > > > Thanks authors for pointing me to their rebuttal to Reviewer wtrT that I missed.
> > > > > > I find their numerical evaluation helpful.  I will update my score from 5 to 6.

---

### Official Review · Reviewer_xjjZ · 2024-11-09

**Soundness:** 3
**Presentation:** 3
**Contribution:** 3
**Rating:** 6
**Confidence:** 3

**Summary:**

This work investigates post training quantization from an accumulator-aware perspective. They aim at using low-precision at accumulation while avoiding the overflow issue. The paper proposes the AXE, as a practical, low-overhead framework as extensions on top of two state-of-the-art PTQ algorithms: GPFQ and OPTQ. The work supports full datapath optimization and scales to large language models. They achieve improvements in the trade-off between accumulator bit width and model accuracy over baseline methods.

**Strengths:**

+ The accumulation-aware approach is well motivated from the hardware and implementation perspective, because when weights and activations are quantized into low-precision, the 32-bit accumulation consumes majority of power and area. And using low-precision on accumulation may increase the risk of numerical overflow which degrades model accuracy.

+ The paper adopts an effective approach to theoretically gurantee overflow avoidance by constraining ||q||1 in post training quantization process. To solve the problem, AXE translate into two accumulator-aware constraints.

+ The multi-stage accumulation extension of AXE is effective in improving throughput and scaling to large language models.

**Weaknesses:**

- The adoption of the two PTQ algorithms GPFQ and OPTQ and the applicability of AXE to other PTQ need justification.

- Because the concept of accumulation aware quantization was proposed from the implementation perspective. It is more convincing to demonstrate the performance in terms of latency or throughput besides model accuracy.

See the questions section for more details.

**Questions:**

The paper designs AXE as extensions on top of existing PTQ algorithms: GPFQ and OPTQ, claiming them as state of the arts. While OPTQ is OK, but GPFQ is an earlier work. Please justify why those two quantization algorithms are picked and comment the applicability of AXE to other PTQ methods.

The accumulation aware approach was motivated from the implementation perspective, but results are only evaluated from the model accuracy performance. Yes, the AXE is effective in avoiding numerical overflow and preserve model performance. Is it possible to justify the accumulation aware approach in terms of latency or throughput in implementation? This can make the work more convincing.

---

> ### Author Response · Authors · 2024-11-19
>
> Thank you for your questions and feedback. To conserve space, we address Q2 in our response to Reviewer wtrT. We appreciate your understanding and hope the following will address your remaining concerns in accepting this paper.
>
> We respectfully emphasize that ours is not a study designed to push the boundaries of the established paradigms of weight-only or weight-activation quantization, but rather the emerging paradigm of accumulator-aware weight quantization. Our intention in designing AXE is to endow any greedy sequential weight-activation PTQ method with accumulator-awareness (please see Section 3.3, lines 353-359). We additionally leverage a collection of other PTQ methods that solve orthogonal problems when establishing our reference baselines without accumulator-awareness (please see Appendix C.1, lines 891-901). It is an artifact of recent naming conventions and benchmarking practices that these quantization methods are perceived as mutually exclusive algorithms when in reality they can be composable and compatible; for example, in their seminal paper on post-training quantization, Nagel et al. (2019) combine "bias correction" (which renormalizes quantization error to zero-mean) and "weight equalization" (which iteratively equalizes the precision of neighboring layers within scale-invariant regions of a model). SmoothQuant, similar to weight equalization, takes advantage of scale invariance to derive a per-channel smoothing factor that balances weight and activation ranges, as opposed to the ranges of weights in neighboring layers. As we focus our analysis on LLMs, we augment our baselines with SmoothQuant and bias correction as they are composable. We respectfully direct your attention to our response to Reviewer wtrT, where we provide new results to demonstrate composability for Llama3 models.
>
> In this work, we present and investigate accumulator-aware variants of both GPFQ and OPTQ to demonstrate the versatility of the AXE framework. We choose GPFQ as it is a principled PTQ technique with provable error bounds and accompanying theoretical analyses (Lybrand & Saab, 2021; Zhang et al., 2023). While our theoretical justification is tied to the formulation of GPFQ and its derivations, we extract our constraints to construct a generalized framework that is compatible with any algorithm that greedily assigns bits element-by-element. Chee et al. (2024) characterize a family of such algorithms, of which OPTQ and GPFQ are two prominent examples. It is a contribution of our work that AXE is compatible with both OPTQ and GPFQ (and other algorithms like them) to offer the first set of principled accumulator-aware PTQ mechanisms that scale to billion-parameter LLMs. For the first time, this enables the co-optimization of large-scale models and their target inference datapaths. Note that we highlight the established benefits of this co-optimization opportunity in Section 2.2 (lines 141-150), Section 3.3 (lines 378-390), and in our response to Reviewer SPYS.

---

### Author Response · Authors · 2024-11-28

We thank the reviewers for their feedback and engagement.

We are encouraged to see that the reviewers acknowledge the unique challenges and opportunities that arise when considering the accumulator in the quantization design space. We are also grateful that the reviewers see the strength in our theoretically principled mechanism for accumulator-aware post-training quantization. We hope our point-by-point responses have addressed any remaining concerns in accepting this paper.

We greatly appreciate the reviewers' efforts in evaluating our work. The constructive suggestions are invaluable for further improving the paper as well as how we communicate our ideas moving forward.

We have revised our paper to include more results with Llama3 in Appendix D.

---

### Meta-Review · Area_Chair_6oVu · 2024-12-13

**Metareview:**

While the problem addressed by the paper—extending post-training quantization (PTQ) to consider accumulator-aware design—is well-motivated, the work falls short in providing sufficient experimental evidence to validate its claims and situate its contributions within the current state of the field.

The primary issue lies in the lack of practical demonstration of the claimed benefits. While the authors propose the AXE framework as a novel method for accumulator-aware PTQ, its evaluation is primarily limited to accuracy improvements on specific LLM models, with limited attention to critical metrics such as latency, throughput, or energy efficiency on real-world hardware. This is a major oversight, given that the core motivation of accumulator-aware quantization lies in resource efficiency. Furthermore, the experiments are constrained to a relatively narrow set of model architectures and do not convincingly demonstrate generalization or practical applicability.

**Additional Comments On Reviewer Discussion:**

During the rebuttal period, reviewers raised concerns about the practical benefits of the proposed method, specifically its impact on latency, throughput, and energy efficiency, as well as the narrow scope of model evaluations. The authors provided additional results with LLaMA models and theoretical insights into the potential efficiency gains but acknowledged the lack of extensive hardware-based validation.

---

### Decision · Program_Chairs · 2025-01-22

Reject